# Freshwater in the Arctic Ocean 2010-2019

Amy Solomon[1,2], Céline Heuzé[3], Benjamin Rabe[4], Sheldon Bacon[5], Laurent Bertino[6], Patrick Heimbach[7], Jun Inoue[8], Doroteaciro Iovino[9], Ruth Mottram[10], Xiangdong Zhang[11], Yevgeny Aksenov[5], Ronan McAdam[9], An Nguyen[7], Roshin P. Raj[6], Han Tang[11]

[1]Cooperative Institute for Research in Environmental Sciences, University of Colorado Boulder, USA
[2]Physical Sciences Laboratory, NOAA Earth System Research Laboratory, Boulder, Colorado, USA
[3]Department of Earth Sciences, University of Gothenburg, Gothenburg, Sweden
[4]Alfred-Wegener-Institut Helmholtz-Zentrum für Polar- und Meeresforschung, Bremerhaven, Germany
[5]National Oceanography Centre, Southampton, UK
[6]Nansen Environmental and Remote Sensing Center and Bjerknes Center for Climate Research, Bergen, Norway
[7]University of Texas, Austin, Texas, USA
[8]National Institute of Polar Research, Tachikawa, Japan
[9]Centro Euro-Mediterraneo per i Cambiamenti Climatici, Bologna, Italy
[10]Danish Meteorological Institute, Copenhagen, Denmark
[11]University of Alaska Fairbanks, Fairbanks, Alaska, USA

*Correspondence to*: Amy Solomon (amy.solomon@noaa.gov)

**Abstract.** The Arctic climate system is rapidly transitioning into a new regime with a reduction in the extent of sea ice, enhanced mixing in the ocean and atmosphere, and thus enhanced coupling within the ocean-ice-atmosphere system; these physical changes are leading to ecosystem changes in the Arctic Ocean. In this review paper, we assess one of the critically important aspects of this new regime, the variability of Arctic freshwater, which plays a fundamental role in the Arctic climate system by impacting ocean stratification and sea ice formation/melt. Liquid and solid freshwater exports also affect the global climate system, notably by impacting the global ocean overturning circulation. We assess how freshwater budgets have changed relative to the 2000-2010 period. We include discussions of processes such as poleward atmospheric moisture transport, runoff from the Greenland Ice Sheet and Arctic glaciers, the role of snow on sea ice, and vertical redistribution. Notably, the sea ice cover has become more seasonal and more mobile, the mass loss of the Greenland Ice Sheet has increased in the 2010s (particularly in the west, north, and south regions), and imported warm, salty Atlantic waters have shoaled. During 2000-2010, the Arctic Oscillation and moisture transport into the Arctic are in-phase and have a positive trend. This cyclonic atmospheric circulation pattern forces reduced freshwater content on the Atlantic-Eurasian side of the Arctic Ocean and freshwater gains in the Beaufort Gyre. We show that the trend in Arctic freshwater content in the 2010s has stabilized relative to the 2000s, potentially due to an increased compensation between a freshening of the Beaufort Gyre and a reduction in freshwater in the rest of the Arctic Ocean. However, large inter-model spread across the ocean reanalyses and uncertainty in the observations used in this study prevent a definitive conclusion about the degree of this compensation.

# 1     Freshwater in the Arctic Ocean

Rapid changes in the Arctic climate system are impacting marine resources and industries, coastal Arctic environments, and large-scale ocean and atmosphere circulations. The Arctic climate system is rapidly transitioning into a new regime with a reduction in the extent of sea ice (Stroeve and Notz, 2018), a thinning of the ice cover (Kwok, 2018), a warming and freshening of the Arctic Ocean (Timmermans and Marshall, 2020), regionally enhanced mixing in the ocean and atmosphere and enhanced coupling within the ocean-ice-atmosphere system (Polyakov et al., 2020); these physical processes are leading to cascading changes in the Arctic Ocean ecosystems (Bluhm et al., 2015; Polyakov et al., 2020). The emergent properties of this new regime, termed the "New Arctic" (Jeffries et al., 2013), are yet to be determined since altered feedback processes are expected to further impact upper ocean heat and freshwater content, atmospheric and oceanic stratification, the interactions between subsurface/intermediate warm waters and surface cold and fresh layer, among other properties (Carmack et al. 2016). In this review we assess one of the critically important aspects of this new regime, the variability of Arctic freshwater.

Freshwater in the Arctic Ocean plays a critical role in the global climate system; by impacting large-scale overturning ocean circulations (Sévellec et al., (2017), see Figure 1 showing basins and upper circulation); by changing ocean stratification that affects sea ice growth, biological primary productivity (Ardyna and Arrigo, 2020; Lewis et al., 2020), and ocean mixing (Aagaard and Carmack, 1989); and by the emergence of freshwater regimes that couple variability in land, atmosphere, and ocean systems (e.g., Jeffries et al., 2013; Wood et al., 2013). Arctic Ocean freshwater is a balance between:

- sources (relatively fresh Pacific oceanic inflow, precipitation, river runoff, ice sheet discharge, sea ice melt) (Aagaard and Woodgate, 2001; Serreze et al., 2006; Bamber et al., 2012),
- sinks (relatively saline Atlantic oceanic inflow, sea ice growth, evaporation, liquid and solid transport through oceanic gateways) (Aagaard and Carmack, 1989; Rudels et al., 1994; Serreze et al., 2006; Haine et al., 2015),
- redistribution between Arctic basins and vertical mixing (e.g., Timmermans et al., 2011; Morison et al., 2012; Proshutinsky et al., 2015).

These processes are not necessarily independent and are largely driven by atmospheric variability both within the Arctic and from lower-latitudes.

Oceanographers have long been accustomed to the use of "freshwater" as an identifiable and separable component of seawater, either as a freshwater volume or a freshwater flux component of a seawater volume or flux. It usually manifests as a small fraction of the seawater volume or flux, where the fraction takes the form ($\delta S/S_{ref}$), and where $\delta S = S{-}S_{ref}$ is the deviation of the seawater salinity $S$ from a reference value $S_{ref}$. The sign in the numerator is conventionally reversed so that a positive scaled salinity anomaly reflects a freshwater reduction, and vice-versa. However, scientists' familiarity with this

usage perhaps disguises the fact that it is an arbitrary construct: the concept of "reference salinity" and values attributed to it are not rigorously mathematically and physically defined.

Since this is a review of existing literature and in light of established practice, we continue to employ here the "traditional" approach to freshwater flux calculation by use of a fixed reference salinity. For completeness we include here a discussion of recent studies that highlight the ambiguity that arises when a constant reference salinity is used to calculate freshwater fluxes. The significant freshwater flux differences that can arise from use of different reference salinities are illustrated and quantified by Tsubouchi et al. (2012), as well as by Schauer and Losch (2019). Schauer & Losch (2019) argue that it is preferable to use the uniquely-defined salt budget as an absolute and well-posed physical quantity. However, Bacon et al. (2015) observed that a true freshwater flux occurs without ambiguity at the surface where freshwater is exchanged between ocean and atmosphere (via precipitation and evaporation) and where the ocean receives freshwater input from the land (as river or other runoff). This atmosphere-ocean surface freshwater flux is a key element of the global freshwater cycle, predicted to amplify with global warming; hence the importance of knowledge of this surface flux, of its impacts on the ocean, and of the ocean's redistribution (and storage) of these impacts. Salinity, by comparison, is of indirect interest for its role in seawater density and buoyancy, etc. Bacon et al. (2015) recognize that a surface flux requires definition of a surface area. They then use a time-varying ice and ocean control volume (or "budget") approach, combined with mass and salt conservation, to generate a closed mathematical expression where the surface freshwater flux is given by the sum of three terms: (i) the divergence of the (scaled) salt flux around the boundary of the control volume, (ii) the change in total (ice and ocean) seawater mass within the control volume (or change in mass storage), and (iii) the (scaled) change in mass of salt within the control volume (the change in salinity storage). The "scaling" term that emerges from the mathematics performs the same function as the traditional reference salinity, but in its place is the control volume's ice and ocean boundary mean salinity, which has uncomfortable implications in that it can vary in time and with boundary geography. This is a consequence of the nature of the calculation, which quantifies surface freshwater fluxes. Carmack et al. (2016) interpret the Arctic case thus: the surface freshwater flux is what is needed to dilute all the ocean inflows to become the outflows, allowing for interior storage changes. An exactly equivalent interpretation is that surface freshwater fluxes and the relatively fresh Bering Strait sea water inflow combine to dilute the relatively saline Atlantic water inflow, which then become the outflows (allowing for storage) – where "relatively" means relative to the boundary mean salinity.

Forryan et al. (2019) pursue the surface freshwater flux approach, noting that (as is well known, e.g., Östlund and Hut, 1984) evaporation and freezing are distillation processes that leave behind a geochemical imprint via oxygen isotope anomalies on the affected freshwater in the sea ice and seawater. In the case of evaporation, distillation (here, isotopic fractionation) preferentially removes lighter oxygen isotopes from seawater, leaving behind in the seawater a proportion of heavier isotopes. The lighter isotopes that are now in the atmosphere return to the land or sea surface as precipitation. Those falling on land can (eventually) transfer from land to sea by river runoff or by other glacial processes, or by further cycles of evapo-

transpiration and precipitation. For sea ice, the ice contains the lighter isotopes while heavier isotopes are contained in the brine that drains out of the ice during freezing, to re-enter the ocean. The isotopically-lighter meteoric fractions are used to quantify freshwater that originates from the atmosphere (directly or indirectly), and the isotopically-heavier fractions similarly quantify the signal of brine rejected from sea ice, and thereby the amount of ice formed from that seawater. The Forryan et al. (2019) study shows that, within uncertainties, the geochemical approach produces the same surface freshwater flux as the budget approach.

Freshwater input to the Arctic Ocean is almost entirely confined to the upper water column and comes in the form of continental runoff, including from glacier melt, waters of Pacific origin, various coastal currents and precipitation. In addition, freshwater input from the Greenland ice sheet and other marine terminating glaciers has three subsurface contributions: (i) melting from calved icebergs (Moon et al., 2017), (ii) submarine melt rates which may produce a freshwater plume, which may or may not become neutrally buoyant below the surface (Straneo et al., 2011), and subglacial runoff. Melt plumes that are amplified by seasonal subglacial runoff, are more likely to reach the surface. Jenkins (2011) refers to a melt plume in the absence of subglacial runoff as "melt-driven convection", whereas runoff (an added buoyancy source) incurs "convection-driven melting". Overall, the freshwater flux magnitude from Greenland into the Arctic Mediterranean remains small compared to that of Arctic river runoff; for the period 1961-1990. Bamber et al. (2012) estimate around 184 $km^3$/yr freshwater flux from North and Northeast Greenland into the Arctic Mediterranean. Given this region is mostly onto the continental shelves adjacent to the Fram and Nares Straits, it is likely that much of the discharge from northern Greenland is rapidly exported. Another 432 $km^3$/yr from West and Southwest Greenland are discharged into Baffin Bay and Labrador Sea, and 266 $km^3$/yr from Southeast Greenland into the Irminger basin. Compare these numbers to a total of 2,440 $km^3$/yr of combined Arctic river runoff over the same period (Bamber et al. 2012); Haine et al. (2015), provide numbers of 3900 $km^3$/yr +/-10% for the period 1980-2000 and 4200 $km^3$/yr +-10% for the period 2000-2010, i.e., almost twice as large).

The upper Arctic Ocean hence is characterized by salinity values lower than that of the inflow of waters of largely Atlantic origin through the Fram Strait and the Barents Sea opening. The result is an extremely stratified Arctic Ocean, with a shallow seasonal mixed layer on average less than 100 m thick and a halocline that is the result of all the inflows (McLaughlin et al., 1996; Rudels et al., 2004). Below the halocline sits the "Atlantic layer", which is comparatively warm and salty, and below this are the Arctic Ocean deep waters (Aagaard et al., 1985; Rudels, 2012). Vertical fluxes of freshwater are generally low due to this strong stratification and very low vertical turbulent mixing / diffusion (e.g., Fer, 2009). The reviews of Carmack et al. (2016) and Haine et al. (2015) confirm the picture above; hence, they mainly considered the Arctic freshwater budget in the near-surface layers. This current study expands on their work and describes the processes impacting the vertical (re)distribution of freshwater throughout the entire water column.

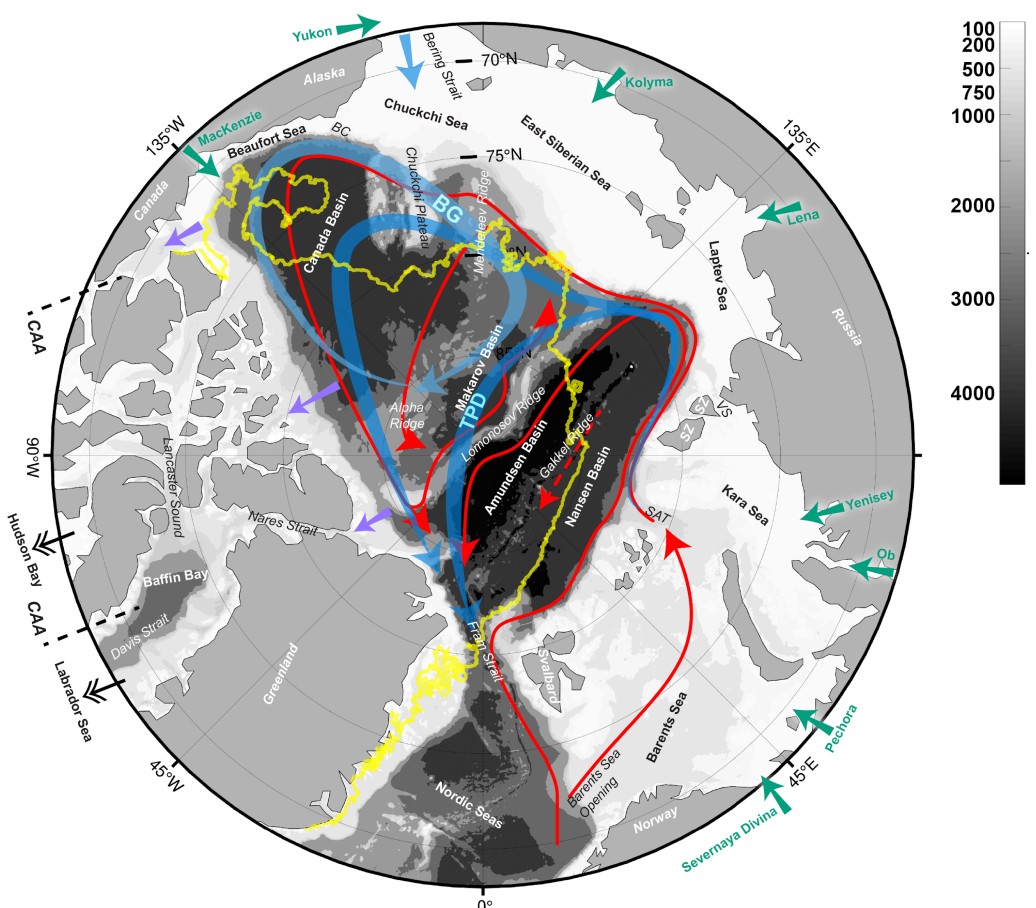

**Figure 1: Map of the Arctic Ocean with names of major basins and shelf seas, and ocean circulation features: major river and Pacific inflow (cyan and turquoise) and surface outflows (purple), 2020 minimum sea ice edge (yellow), cold and fresh upper ocean circulations (Polar Surface Water and halocline, blues), and warm and salty Atlantic water circulation (red). Areas shallower than 1000 m are referred to as shelf areas in the text. BG stands for Beaufort Gyre; TPD, Trans-polar drift; BC, Barrow Canyon; CAA, Canadian archipelago; SAT, St Anna Trough; VS, Vilkiltskiy Strait; SZ, Severnaya Zemlya.**

Assessments of Arctic freshwater for the 2000-2010 period relative to 1980-2000 were completed as part of the WCRP/IASC/AMAP Arctic Freshwater Synthesis (Prowse et al., 2015; Carmack et al., 2016; Vihma et al., 2015) and the Arctic-Subarctic Ocean Fluxes program (Haine et al., 2015). These projects found that liquid freshwater increased by 25% (5000 km³) in the Beaufort Gyre; the Beaufort High was stronger than normal with higher sea level, a deeper halocline, stronger anticyclonic flow, and stronger Trans-polar Drift (Proshutinsky et al., 2009; McPhee et al., 2009; Rabe et al., 2011; Haine et al., 2015). However, estimates of fluxes through the Fram Strait and the Labrador Sea were either too uncertain or

showing statistically insignificant changes, leading to speculation whether freshwater accumulated in the Arctic Ocean, if released via these Arctic gateways, could substantially impact the global ocean overturning circulation and climate (e.g., Haine, 2020; Zhang et al., 2021). In these studies, processes such as the redistribution of freshwater between basins and vertical redistribution due to turbulent mixing were not taken into account, leading to uncertainty in this speculation.

The observed Beaufort Gyre freshening is illustrated in Figure 2, which shows 1993-2019 annual mean Arctic Ocean freshwater from seven state-of-the-art global ocean reanalyses (ORAs, see Table 1 for a description of the models used in this study). Significant freshening in the Beaufort Gyre is seen in 2010-2017 means minus 2000-2010 means in six ORAs (Figure 2b, not including ASTE_R1, using the common period of 2010-2017 for the difference maps). However, this freshening is partly compensated by a reduction in freshwater in the rest of the Arctic Ocean (Figure 2b,c). This compensation increases in 2010-2018 compared to 2000-2010, which flattens the total Arctic Ocean freshwater trend when extended to 2019 (Figure 2a). This is characteristic of the cyclonic mode of circulation (Morison et al., 2021; Morison et al., 2012; Sokolov, 1962). However, there is a significant spread in estimates of freshwater content in the Beaufort Gyre and the rest of the Arctic Ocean (Figure 2d), which prevents a definitive estimate of the degree of this compensation. The wide variability in FWC change among the ORAs in the Siberian Shelf Seas is likely due to the paucity of observations there in recent years. Morison et al. (2021) speculate that the in-situ observations have had an increasing spatial bias toward the Beaufort Sea. This highlights the need to be able to estimate the redistribution of freshwater when assessing changes in Arctic Ocean freshwater, as well as the recent reduction in total Arctic Ocean freshening relative to the 2000-2010 period.

In this review we assess to what extent the 2010-2019 freshwater budget has changed relative to the 2000-2010 period. This study is not meant to be a comprehensive assessment of all processes that contribute to Arctic freshwater. Instead, we focus on specific aspects that provide insight into how the variability has changed since 2010 and the role of processes not considered in previous assessments.

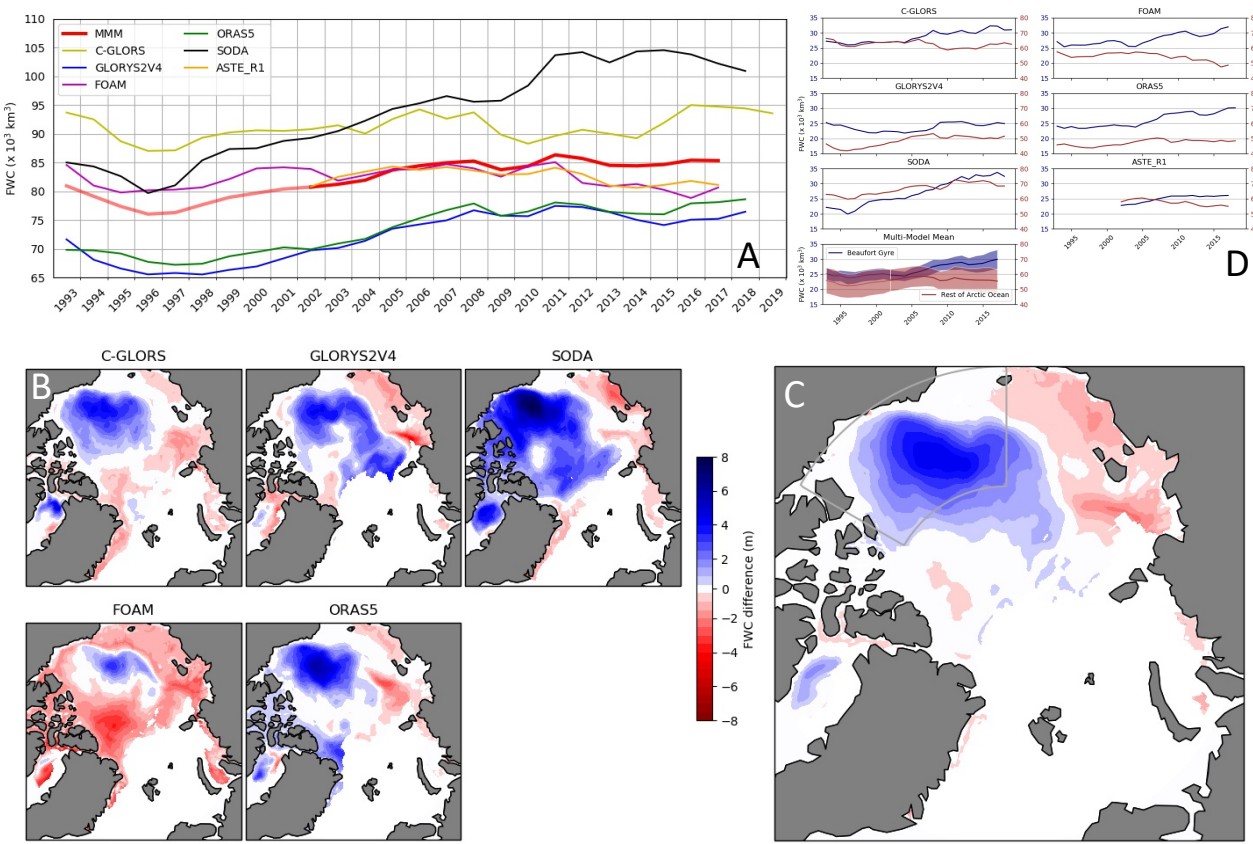

Figure 2: Ordered counter-clockwise: A) Time series of annual freshwater content integrated from 70-90°N and down to the 34 isohaline for the period 1993-2019 from 6 ORAs (in 10³ km³). Multi-model mean shown in red, darker red indicates all 6 ORAs included. B) Difference (in m) between 2010-2017 and 2000-2010 means in 5 ORAs (not including ASTE_R1). C) Multi-model mean of differences (in m) shown in (B). D) Annual freshwater content separated into contributions from Beaufort Gyre (blue) and the rest of the Arctic Ocean (red). Lower right figure shows the multi-model mean with +/- 1 standard deviation shown with shading. The Beaufort Gyre is defined as 70-80°N, 120-180°W to be consistent with the satellite estimates below.

| Name | C-GLORSv7 | FOAM | GLORYS2V4 | ORAS5 | ASRE_R1 | SODA3.3.2 |
|---|---|---|---|---|---|---|
| **Institution** | CMCC | UK MetOffice | CMEMS | ECMWF | Univ. Texas Austin | Univ. Maryland |
| **Horizontal Resolution** | 0.25° | 0.25° | 0.25° | 0.25° | 0.3° | 0.25° |
| **Vertical Resolution** | 75 z-levels | 75 z-levels | 75 z-levels | 50 z-levels | 50 z-levels | 50 z-levels |
| **Surface Fluxes** | CORE | CORE | CORE | CORE+ Wave Forcing | CORE | COARE4 |
| **Atmospheric Forcing** | ERA-Interim | ERA-Interim | ERA-Interim | ERA-Interim until 2014 ECMWF NWP after 2014 | Adjusted JRA55 | MERRA2 |
| **Ocean-Sea Ice Model** | NEMO3.2-LIM2 | NEMO3.2-CICE | NEMO3.1-LIM2 | NEMO3.2-LIM2 | MITgcm | MOM5-SIS1 |
| **DA Variables** | SIC, Arctic SIT, T, S, SSH, SST | SIC, T, S, SST, SSH | SIC, T, S, SST, SSH, runoff | SIC, T, S, SST, SSH | SIC, T, S, SST, SSH | T, S, SST, Greenland and river runoff |
| **DA Sources** | OIv2d, PIOMAS, EN4, AVISO | OSISAFv2, EN4, ICOADS, AVHRR,ATSR, AMSRE, AVISOv3 | CERSAT, CMEMS, AVHRR | OSTIA, Olv2d, EN4, AVISO | ARGO, ITP, ICES, XBT, CTD | WOD, ICOADS, AVHRR, Metosat, SEVIRI |
| **References** | Storto & Masina (2016) | Blockley et al. (2014) | Garric et al. (2017) | Zuo et al. (2019) | Nguyen et al. (2021) | Carton et al. (2018) |

**Table 1: Global ocean reanalyses used in this study**. T temperature data, S salinity, SST sea surface temperature, SSS sea surface salinity, SSH sea surface height, SIC sea-ice concentration, SIT sea-ice thickness, DA data assimilation, CORE Coordinated Ocean-Ice Reference Experiment, COARE4 Coupled-Ocean Atmosphere Response Experiment bulk flux parameterization version 4.

## 1.1 Arctic freshwater estimates from in-situ and satellite measurements

### 1.1.1 Satellite measurements

A major challenge in the retrieval of freshwater fluxes in the Arctic Ocean is associated with the lack of availability of in-situ observations. Direct measurements are non-homogenous in both time and space and rely on spatial as well as temporal interpolation resulting in large uncertainties. The ability to estimate freshwater content of the Arctic region indirectly from satellite observations is a major breakthrough. The methodology which exploits the satellite derived ocean mass change and satellite altimeter data is detailed in Giles et al. (2012), Morison et al. (2012), and Armitage et al. (2016). It derives from the perceptions (1) that the sea surface height change (as observed by satellite altimeters) is the sum of two components: mass addition (or loss) and steric expansion (or contraction); and (2) that observation of mass changes (by satellite gravimetry) enables separation of these two components. A two-layer model is assumed, where the sea surface height and interface depth are variable, where the upper layer represents the halocline (and surface mixed layer) and the lower layer all underlying waters. Upper layer thickness changes (per unit water column area) are then a function of changes in sea surface height and water column mass, with assumed layer densities; and changes in freshwater content are then the thickness changes scaled by ($S/S_{ref}$), with reversed sign. Giles et al. (2012) assume $S_{ref}$ (their $S_2$) equal to 34.7 and upper layer salinity 27.7. While this is a very simple model, the observed signals are significantly larger than the uncertainty, as shown in their thorough uncertainty assessment (Giles et al. 2012 Supplementary Information).

Our understanding of the Earth's gravity field has improved considerably during the recent decade, thanks to the Gravity Recovery and Climate Experiment (GRACE) mission launched in 2002. GRACE is the only satellite mission designed to be directly sensitive to mass changes by means of gravity. The variability in spatiotemporal characteristics of the Earth's gravitational field resulting in very small deviations in the separation between the two satellites of the GRACE mission are measured with micrometer precision and are used to infer the Earth's gravity field, which can then be used to estimate changes in ocean mass (Peralta-Ferriz et al., 2014; Armitage et al., 2016). Here we use the latest Release-06 gridded GRACE ocean mass products from the Jet Propulsion Laboratory (Watkins et al., 2015). Satellite radar altimeters on the other hand can retrieve sea surface heights in the open ocean with variable precision depending on the number of flying altimeters and have been uninterrupted since 1993. CryoSat-2, launched in 2010, is a satellite altimeter that provides coverage up to 88°N with much better spatial resolution than before. Several studies have utilized this source to study the sea level variability of the Arctic (Kwok and Morison, 2016, 2017; Armitage et al., 2018a, 2018b; Rose et al., 2019; Raj et al., 2020). However, constructing precise altimeter derived sea level data in the Arctic Ocean is still a challenge, one of them is the effect of melt ponds during summer on the waveforms which dominate the reflected signal. A better understanding of the radar altimeter response over the different ice types must be gained to improve the quantity and quality of the range retrievals in the Arctic Ocean. One of the ongoing efforts is the currently ongoing CRYO-TEMPO project, funded by the European Space Agency.

Satellites can monitor some important pieces of the Arctic freshwater puzzle. Here, we use the state-of-the-art sea level product produced as part of the recently concluded climate change initiative (CCI) project (Sea level budget closure; Horwath et al., 2020) of the European Space Agency. This Arctic sea level product (DTU/TUM SLA record; Rose et al., 2019) is the first one which includes a physical retracker (ALES+) for retrieving the specular waveforms from open leads in the sea cover. The sea state bias corrected using ALES+ improves the sea level estimates of the region (Passaro et al., 2018). The latest version (v3.1) of the DTU/TUM SLA record is a complete reprocessing of the former DTU Arctic sea level product (Andersen et al., 2016) by dedicated Arctic retracking. The current study thus takes advantage of the state-of-the-art satellite datasets to study the freshwater content of the region following Giles et al. (2012) and Armitage et al. (2016).

Freshwater is calculated using the satellite measurements using these equations:

$$\Delta \text{FWC} = \frac{S_2 - S_1}{S_2} A \sum_{i=0}^{N} \Delta h_i, \tag{1}$$

$$\Delta h = \eta \left(1 + \frac{\rho_1}{\rho_2 - \rho_1}\right) - \frac{\Delta m}{\rho_2 - \rho_1}, \tag{2}$$

Where $\eta$ is the change in SSH, $\Delta m$ is the ocean mass anomaly, N is the number of grid cells and A is the grid cell area. The salinities S1, and S2 are respectively 27.7 and 35, while densities $\rho 1$ and $\rho 2$ are 1022 and 1028 respectively, in units of kg m$^{-3}$.

Time series from 2002 to 2018 using GRACE-derived ocean bottom pressure (OBP) anomalies (https://podaac.jpl.nasa.gov/GRACE) and satellite altimeter data provide insights into the redistribution of freshwater in the Arctic Ocean (Figure 3). While initial results from GRACE suggest an overall OBP decrease caused by a fresher Arctic surface (Morison et al., 2007), results on the now-longer time series show more complex interannual variability, in agreement with modelling data (e.g., de Boer et al., 2018). Figure 3a (red line) shows that freshwater content increases in the Beaufort Gyre during the time-period 2002-2010, followed by a stabilizing phase where the increase flattens out. However, including freshwater content outside of the Beaufort Gyre (blue line in Figure 3a, defined as the region contoured in Figure 3c) results in a reduction in freshwater content during the time-period 2010-2016, indicating increased compensation between freshwater content in the Beaufort Gyre and outside the Beaufort Gyre after 2009. Raj et al. (2020) noted a similar signature in the altimeter derived sea surface height anomaly and the halosteric component of the sea surface height anomaly and attributed it to the change in the dominant atmospheric forcing over the Arctic, which changed from the Arctic dipole pattern to the Arctic Oscillation respectively during the time-periods prior-to and after 2010. These results are qualitatively consistent with estimates in Figure 2 using the ocean reanalyses. In addition, Figure 2c shows that the regions not included in Figure 3 make only small contributions to the time series In Figure 2. It is well-known that, while the sea surface height variability in

the Beaufort Gyre region is dictated by the variability in salinity, the same in the Nordic Seas and the Barents Sea is controlled by Atlantic Water temperature in comparison to salinity (Raj et al., 2020). Hence the methodology to estimate FWC from sea surface height data is not recommended in those two regions. Our study included the rest of the Arctic excluding the Canadian Archipelago, Nordic and the Barents Sea.

### 1.1.2    In-situ measurements

Figure 3b includes estimates of Arctic freshwater content from in-situ hydrographic observations (black line). The timeseries of freshwater content for the whole basin to the 34 isohaline is extended from Rabe et al. (2014a). Details of the mapping procedure and the distribution of hydrographic stations until 2012 are given in Rabe et al. (2014a). Further data is based on the data sources listed in Table 2. Interestingly, the Arctic satellite and in-situ time series in Figure 3a,b are relatively consistent before 2009, but do not show the same variability after 2009. This difference may stem from the lack of data coverage in the in-situ measurements, the different regions used in the time series and the choice of time period for the mean used to obtain anomalies. The satellite time series uses the region contoured in Figure 3c and the in-situ time series uses observations within the basin excluding the shelves, indicating a good part of the difference after 2009 may be due to the contribution by the rest of the basin outside the Beaufort Gyre. In addition, the annual values of the in-situ time series are biased towards the prior three years near the end of the timeseries, as the mapping analysis only includes data up to 2015; and 2012, 2013 and 2014 show similar levels as 2015. The locations of all profiles used between 1992 and 2015 show that there are interannual variations in data coverage, but that overall, the decadal timescale is reasonably well covered across the Arctic Ocean basin (Figure 3).

240

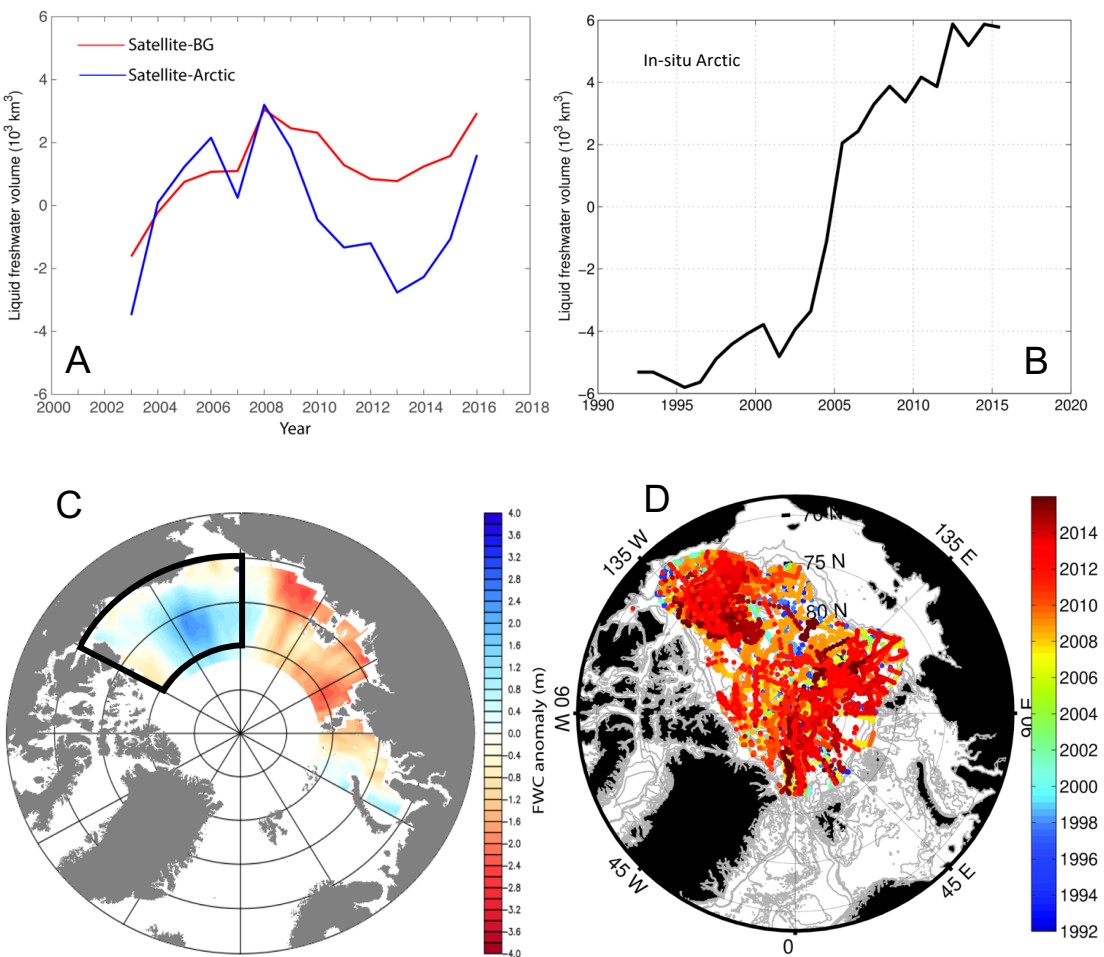

250

**Figure 3: Anomalies of freshwater content from satellite sea-surface height data analysis and GRACE OBP data and from objectively mapped in-situ hydrographic observations. Annual mean time series of freshwater content from (A) satellite measurements in the Beaufort Gyre (red) and Arctic region shown in (C) (blue), and (B) Arctic basin using in-situ hydrographic observations shown in (D) (black), in units of $10^3$ km³. (C) Difference between 2010-2017 and 2002-2010 freshwater content means from satellite measurements, in units of meters. (D) Locations of salinity profiles used for the objective analysis of the in-situ data with time denoted by color. Anomalies in (A,B) are relative to the corresponding mean of the period 2003-2006 in each time series using a reference practical salinity of 35 and a layer from the surface to the 34 isohaline. The Beaufort Gyre region (marked with thick black lines in (C)) is defined as 70-**

260 **80°N, 120-180°W. The time series are calculated using observations from the Arctic Ocean with a water depth deeper than 500 m and a cut-off at 82°N north of the Fram Strait for the in-situ estimates, and the contoured region shown in (C) for the satellite estimates. (B) is an update of the time series in Rabe et al. (2014a), partly shown previously in Wang et al (2019), the additional data used is listed in Table 2.**

| Expedition, Project | Year(s) | Platform | Source URL or contact |
|---|---|---|---|
| Beaufort Gyre Project | 2012-2013 | various ships | http://www.whoi.edu/beaufortgyre/ |
| NPEO | 2012-2014 | Airborne and ice-based | ftp://psc.apl.washington.edu/NPEO Data Archive/NPEO Aerial CTDs/ |
| WHOI | 2012-2015 | ITP | https://dx.doi.org/10.1029/2009JC005660 (Toole et al., 2016) |
| PS86 | 2014 | RV Polarstern | http://doi.pangaea.de/10.1594/PANGAEA.853768 (Vogt et al., 2015) |
| PS87 | 2014 | RV Polarstern | http://doi.pangaea.de/10.1594/PANGAEA.853770 (Roloff et al., 2015) |
| PS94 | 2015 | RV Polarstern | https://doi.org/10.1594/PANGAEA.859558 (Rabe et al., 2016) |
| NABOS | 2013 | NABOS | https://uaf-iarc.org/nabos/ (Polyakov et al., 2003) |

**Table 2: Sources of salinity data used in the objective analysis to derive the black curve in Figure 3. The listed data sources are for the data used in addition to the data described in Rabe et al. (2014a) and published in Rabe et al. (2014b). Abbreviations: ITP -- Ice-Tethered Profiler, NPEO -- North Pole Environmental Observatory, WHOI -- Woods Hole Oceanographic Institution, NABOS -- Nansen and Amundsen Basin Observational System.**

## 2    Changes in Arctic Freshwater Sources and Sinks

The most recent estimates of Arctic freshwater sources and sinks have been developed by Østerhus et al. (2019), Haine et al. (2015), Prowse et al. (2015), Carmack et al. (2016), and Vihma et al. (2016). Only Østerhus et al. (2019) covers a more recent period through 2015. One issue is that not all these estimates use the same reference salinity; a discussion of freshwater versus salt transports and reference salinities is provided in Bacon et al. (2015), Schauer and Losch (2019), and Tsubouchi et al. (2018). Another more recent development over the last decade is the inclusion of freshwater fluxes from the Greenland Ice Sheet (GIS) and smaller Arctic glaciers and ice caps (GICs) into these basins (Bamber et al., 2012; Bamber et al., 2018; Dukhovskoy et al., 2019).

## 2.1    River Discharge

Observations suggest a linkage between the Arctic Oscillation (AO) and the North American (mainly Mackenzie River) runoff pathways (Yamamoto-Kawai et al., 2009; Fichot et al., 2013). There has been a shift from a rather direct outflow via the Canadian Arctic Archipelago (CAA) in the early 2000s to a northward pathway into the Beaufort Gyre around 2006, coinciding with a change to a strongly positive AO. In addition, for high AO indices, river runoff entering the Eurasian shelves is mainly transported into the Canada Basin, while for low AO indices, the transport is mainly towards the Fram Strait by a strengthened transpolar drift (Morison et al., 2012; Alkire et al., 2015).

Observations of runoff rates for Eurasian rivers are available since 1936, and for North American rivers since 1964 (Shiklomanov et al., 2021). There has been a decline since about 1990 in the total gauged area, by ~10%, in Siberia and Canada (Shiklomanov et al., 2021), due to the closure or mothballing of gauging stations.  Regardless, only the most important rivers are gauged:  knowledge of net (continent-scale) river discharge rates requires estimation of the substantial ungauged runoff fraction, typically one third of the total. The long-term, multi-decadal, gauged annual mean runoff rates are given by Shiklomanov et al. (2021) as 1800 km$^3$ yr$^{-1}$ (Eurasia, 1936-2015) and 1150 km$^3$ yr$^{-1}$ (North America, 1964-2015), for a total of 2950 km$^3$ yr$^{-1}$.  Shiklomanov et al. (2021) also note the increase (with uncertainties) in these records as 2.9±0.4 (Eurasia, using 1935-2015 period) and 0.7±0.3 (North America, using 1964-2015 period) km$^3$ yr$^{-2}$. The significant Eurasian trend is of order 15% per century.  However, the weakly-significant North American trend over the shorter period disguises an apparent signal of multi-decadal variability similar to that observed by Florindo-Lopez et al. (2020), who suggest it to be part of the evidence for much wider-area atmospheric and oceanic teleconnections.

## 2.2    Precipitation and Atmospheric Moisture Transport

Precipitation over the Arctic is the main source of freshwater into the Arctic Ocean, when including that from its pathway through river discharge from the large continental drainage basins (Haine et al., 2015; Serreze et al., 2006). In climatology, river discharge is predominantly from precipitation though land surface processes (e.g., thawing permafrost, decreasing vegetation transpiration) may have slight contributions according to Zhang et al. (2013). The total continental runoff into the Arctic Ocean is about 0.1 Sv (see Table 1, Haine et al., 2015). The remaining sources are lower, those of similar order of magnitude are Precipitation-Evaporation and Bering Strait liquid inflow. In addition, precipitation is largely driven by atmospheric moisture transport.  Based on a mass-corrected atmospheric moisture transport dataset, Zhang et al. (2013) found that the observed increase in the Eurasian Arctic river discharge was driven by an enhanced poleward atmospheric moisture transport into the river basins.  Using the same dataset, Villamil-Otero et al. (2017) also found a continual enhancement of the poleward atmospheric moisture transport across 60ºN into the Arctic Ocean from the 1950s to the mid 2010s. An update of the transport using ERA5 reanalysis shows a continuation of the enhancement across 60ºN (Figure 4). Nygard et al. (2020) also found an increase in poleward moisture transport from 1979-2018 using the ERA5 data.  Interestingly, they also found that evaporation shows a negative trend due to suppression by the horizontal moisture transport.

The large-scale atmospheric circulation may play a dynamic driving role in the enhanced atmospheric moisture transport. A statistical analysis indicates a temporally-varying relationship between the annual moisture transport and the annual mean Arctic Oscillation (AO, Thompson and Wallace, 1998), showing a negative and a positive correlation before and after 2000. The positive phase of the AO indicates a strengthening of the westerlies, transporting atmospheric moisture to the Eurasian continent and leading to an increase in precipitation over the landmass (e.g., Kryzhov and Gorelits, 2015). An AO positive trend occurred primarily from the late 1980s to mid 1990s. In the 1990s, the variability of the atmospheric circulation was mainly characterized by the AO. However, although a positive correlation occurred between the transport and AO after 2000, the AO lagged the transport variability by one year during the negative peaks in 2005 and 2010. During 2000-2010, the AO mainly showed fluctuations and was also inclined towards a negative phase (Figure 4). After 2010, the AO and transport are in-phase, with a positive trend and peak positive values on 2011 and 2015.

During the 2000-2010 time period, the atmospheric circulation spatial pattern has experienced a radical change in particular during winter seasons, as revealed in Zhang et al. (2008). This changed spatial pattern, named the Arctic Rapid change Pattern (ARP), exhibits a predominant role in driving the poleward moisture transport (Zhang et al., 2013). This driving role can also be manifested by a poleward extension and intensification of the Icelandic Low in the negative ARP phase. Considering that temporal-varying features of AO and the seasonal preference of the emergence of the spatially transformed ARP, the dynamic driving role of the atmospheric circulation needs to be further investigated. In addition, synoptic-scale analysis also suggested the propagation of intense storms into the Arctic played an important role in the enhanced poleward moisture transport and resulting increase in precipitation (e.g., Villamil-Otero et al., 2017; Webster et al., 2019).

Much of the precipitation in the Arctic falls as snow but projections show an increasing amount of rain as the climate warms (Bintanja 2018). This appears to have been tentatively observed in Greenland, though mostly in southern and western Greenland away from the central Arctic Ocean (Doyle et al., 2015; Haine et al., 2015; Boisvert et al., 2018; Oltsmanns et al., 2019), where consequences for surface melt, surface runoff, and ice dynamics from increased rainfall over the ice sheet have been observed (e.g., Lenaerts et al., 2019). Similarly, Webster et al. (2019) note an increased frequency of rain on sea ice. Unfortunately, precipitation is notoriously difficult to measure, particularly in the solid phase, and as with other observations in the Arctic, reliable observations of precipitation are few and far between. Estimates of the precipitation flux are therefore forced to rely on model reanalysis, which have large uncertainties (e.g., Bromwich et al., 2018), on indirect measures such as river runoff, which may also be affected by glacier melt or on GNSS data analysis of solid earth movements in response to localized precipitation (e.g., Bevis et al., 2019).

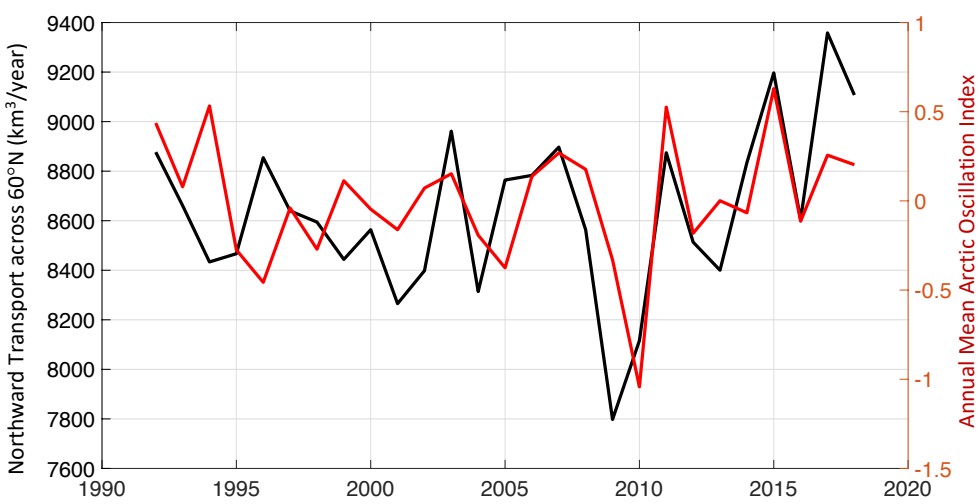

**Figure 4: Time series of annual poleward atmospheric moisture transport (in km³ yr⁻¹) across 60°N updated using ERA5 reanalysis dataset following Zhang et al. (2013) and the annual mean Arctic Oscillation (AO) Index constructed by NOAA Climate Prediction Center from 1991-2019. The transport was integrated from surface to the top of the atmosphere and along 60°N.**

## 2.3    Sea Ice

Freshwater stored in sea ice, i.e., sea ice volume, decreased by roughly 10% for maximum sea ice and 40% for minimum sea ice over 2000-2010 (Figure 5). Kwok (2018) explained the flattening by the predominance of seasonal ice. Using a different approach, Liu et al. (2020) converted sea ice age into volume and also found a decrease in sea ice volume over the entire Arctic of −411 km³ yr⁻¹ over 1984-2018, most pronounced until 2010; their monthly trend ranges between −537 km³ yr⁻¹ in May and −251 km³ yr⁻¹ in September. The decrease in sea ice thickness is responsible for 80% of this trend in winter and 50% in summer. In addition to the global warming effects, sea ice decrease can be attributed to increased downward radiative forcing and turbulent heat fluxes associated with changes in the atmospheric circulation. In particular, storm activities have been intensified over the Arctic Ocean. Recent observational studies have indicated that storms can increase mixing between surface cold water and underlying warm water to suppress winter sea ice growth or increase summer sea ice melt (Graham et al., 2019; Peng et al., 2021). Further, even in the deep basin area where the Pacific and Atlantic waters layers are deeper and stratification is strong, intense storm can force Ekman upwelling to cause the intrusions of the deeper warmer and saltier waters in the upper mixed layer. The input of deep warm and salty waters and enhanced mixing in the mixed layer increase the oceanic heat flux and consequently accelerates summer sea ice melt (Graham et al. 2019; Peng et al., 2021; Polyakov et al., 2020). These processes influence both, the volume of solid freshwater stored in sea ice and ocean freshwater budgets.

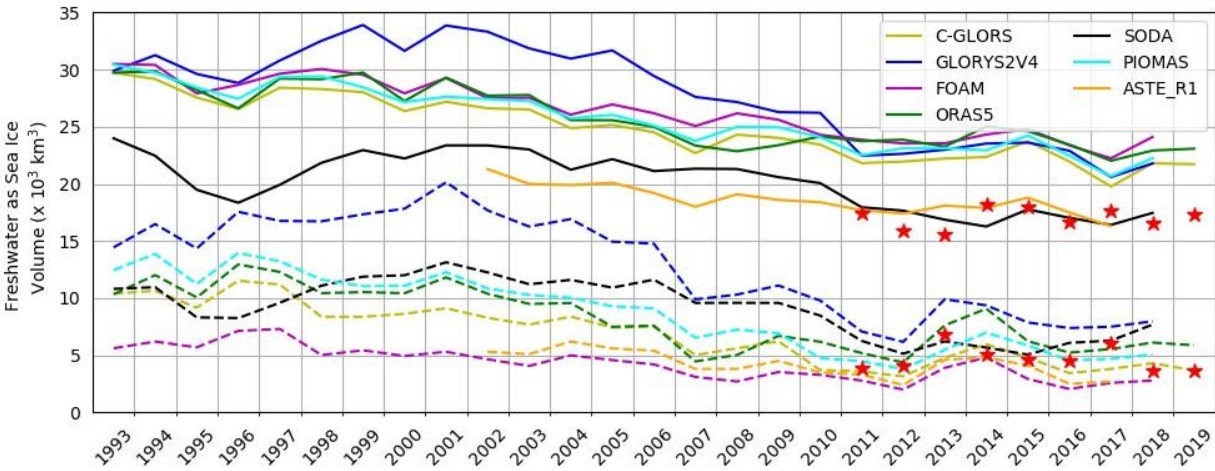

Figure 5: Time series of annual freshwater volume stored as sea ice from 7 ORAs and CRYOSAT2 (red stars), (in $10^3$ km³). The sea ice volume is calculated as the product of sea ice area and thickness. Annual volume maxima are shown by bold lines, while annual minima are shown by dashed lines.

New sea ice in the Arctic forms predominantly over the continental shelf. Estimates based on satellite imagery puts the cumulative sea ice formation of all Arctic coastal polynyas to 3000 km³ per year (Tamura and Oshima, 2011), i.e., about a quarter of the total mean Arctic sea ice volume. Consequently, although the shelves receive large amounts of freshwater from rivers, their largest contribution to freshwater exchanges comes from sea ice export (e.g., Volkov et al., 2020), as the sea ice that forms on the shelves does not stay there. Sea ice is instead slowly transported across the Arctic by the Trans-polar Drift (Serreze et al., 1989), taking 1.5 to 3 years to travel from the Laptev Sea to Fram Strait (Pfirman et al., 1997; Steele et al., 2004). The Trans-polar Drift and ice deformation rates have been observed to be accelerating since the early 2000's (Rampal et al., 2011; Spreen et al., 2011); just recently, the MOSAiC drift expedition (https://mosaic-expedition.org/; Krumpen et al., 2020) has shown that the Trans-polar Drift can, indeed, be unusually fast, thus capable of rapidly transporting sea ice out of the Arctic.

Fram Strait sea ice export is the largest dynamic sink of the Arctic freshwater cycle. The increase in Fram Strait sea ice export detected from long-term monitoring of sea ice area has been suspected as the cause of Arctic sea ice volume loss, in particular for the multiyear thick sea ice within the Arctic Ocean (Smedsrud et al., 2017; Ricker et al., 2018). Using the more recent sea ice thickness retrievals, Spreen et al. (2020) actually showed that in volume, the Fram Strait export has in fact been decreasing at 27% per decade over 1992-2014, on par with the Fram Strait and Arctic ice thickness. In addition to the changes caused by thinned sea ice, changes in the atmospheric circulation pattern have also significantly contributed to the decrease in Fram Strait sea ice export since the mid 1990s (Wei et al., 2019). Sea ice export from the Siberian shelf has

increased by 46% over 2000-2014 compared to 1988-1999; from Amerasia to Europe, by 37% (Newton et al., 2017). But the summer survival rate of sea ice on the Siberian shelves is decreasing by 15% per decade (Krumpen et al., 2019). That is, in the 1990s, 50% of first year ice entered the Trans-polar Drift; now, it is less than 20%, as the rest melts before reaching the Trans-polar Drift (Spall, 2019; Krumpen et al., 2019).

Snow on sea ice is crucial for surface heat budgets through its high albedo, and sea-ice growth through its thermal insulating effect. Therefore, snow on sea ice plays a significant role in determining where and when sea ice melts (Bigdeli et al., 2020). Although the delay of freeze up during early winter, partly depending on the anomalies of oceanic and atmospheric circulations (e.g., Kodaira et al., 2020), would cause a delay of snow accumulation on sea ice, the increase in precipitation and snow depth associated with the increase in storm activities in the Pacific Arctic contributes to a rapid build-up of snow cover on first year ice (and a potential delay in spring/summer sea ice melt). These feedbacks were reported by Sato and Inoue (2018) based on the analysis of Ice Mass Balance buoys and CFSR reanalysis data sets. In the Atlantic sector, precipitation associated with six major storm events in 2014/2015 during the N-ICE2015 field campaign (Merkouriadi et al., 2017) caused the snow depth to be substantially greater than climatology.

## 2.4 Freshwater flux from glaciers and the Greenland Ice Sheet

The freshwater input from Arctic glaciers and the Greenland ice sheet comprises a minor but difficult to compute source of freshwater in the Arctic Ocean basin. The Greenland Ice Sheet and the surrounding smaller peripheral glaciers and ice caps on Greenland have shown an increasing tendency for net ice sheet loss since the early 2000s (Shepherd et al., 2020; Noel et al., 2017; Bolch et al., 2013), though with wide spatial and large temporal variability from year to year, a trend reflected in other glaciated basins within the Arctic including Arctic Canada, Russia and Svalbard (e.g. Noel et al., 2018; Gardner et al., 2011; Moholdt et al.; 2012, Noel et al., 2018). The Ice sheet Mass Balance Intercomparison Exercise (IMBIE) (Shepherd et al., 2012) and IMBIE2 (Shepherd et al., 2020) results for example, show a steady increase in net mass loss from around −119 ± 16 Gt yr[-1] for the period 1992–2011, to −244 ± 28 Gt yr[-1] in the 2012 to 2017 period with a peak in 2012 of 345 ± 66 Gt yr[-1] (see also Helm et al., 2014). The increase in ice loss is due to both enhanced calving and submarine melting at outlet glaciers and increased surface melt and runoff through the period. In the mid-2010s a series of cooler summers, wetter winters and a slowing in calving rates from some of the very large calving outlet glaciers around Greenland led to a short-lived slowing in the rate of mass loss. Simonsen et al. (2021) found that 2017 is the first year in the 21[st] century with a neutral annual mass budget. However, they and others also further note the resumption of high ice loss in 2018 and particularly 2019, which although outside the IMBIE2 period of mass change has led to further decreases in the decadal mass balance of the ice sheet (Tedesco and Fettweis, 2020; Sasgen et al., 2020). However, much of the runoff and solid discharge is lost to the north Atlantic rather than the central Arctic directly and it remains a difficult contribution to estimate accurately. Net ice loss refers to the total mass budget of glaciers where ablation and calving losses exceed gains due to precipitation of primarily snowfall, and a more minor contribution from rainfall that freezes internally within the surface snowpack. Total freshwater flux from

glaciers is consequently rather larger than net ice loss. The main mechanisms of ice loss are: 1) liquid meltwater runoff from both surface and basal melting at the bed of glaciers 2) submarine melt at outlet glaciers in contact with the ocean, 3) a solid component of ice loss driven by the calving of icebergs. All components of ice loss have seen recent increases (Shepherd et al., 2020).

Given the lack of streamflow measurements in Greenland, calculation of liquid runoff is primarily based on numerical models. Meltwater production is calculated within models, based either on surface energy budget considerations or using temperature index scaling, and then runoff is determined by also accounting for refreezing or storage of meltwater in the snowpack. Recent model intercomparisons of modelled Surface Mass Budget (SMB) (Fettweis et al., 2020) and refreezing in firn (Vandecrux et al., 2020) show that the primary source of variability in model estimates is still the amount of melt. This is primarily modulated by surface albedo, but is also determined by the amount and spatial variability in the distribution of snowfall from models as the difference in surface properties between fresh snow and bare glacier ice lead to a melt-albedo feedback that is triggered when bare glacier ice is exposed (e.g., Hermann et al., 2018). The GrIS SMBMIP (Fettweis et al., 2020) compared results from 13 different models over Greenland. While many of these models give a similar figure for the net SMB over the ice sheet, there were wider differences between the components and also the distribution of melt and runoff. Typical values for the mean annual snowfall are in the range of 500 to 800 Gt yr$^{-1}$. The modelled liquid runoff by comparison is in the range of 200 to 500 Gt yr$^{-1}$, though note that many of the highest snowfall models also have runoff so the models converge to a smaller range of SMB values.

To assess the calving and submarine melting components of freshwater flux from Greenland, remote sensing observations have focused on two separate techniques. The discharge method produces an estimate based on the observed velocity of outlet glaciers through flux gates of a known channel cross-section, thus including both solid and liquid ice loss components. The gravimetry method on the other hand estimates mass change over a given area and time period computed from gravimetric observations using the GRACE and later GRACE-Follow On satellites. Modelled SMB is subtracted from the total mass change to give an estimate of the dynamical discharge component that also includes submarine melting at glacier fronts.

Mankoff et al. (2019) used the discharge technique in a recent assessment of the freshwater flux from Greenland to estimate a flux of 488 +/- 49 Gt yr$^{-1}$ that is consistent with that produced by King et al. (2018) of 484 +/-9 Gt yr-1 and Kjeldsen et al. (2015) of $-465.2 \pm 65.5$ Gt yr$^{-1}$, both for the 2003-2010 period. All three studies note that while the amount of discharge over the whole ice sheet has steadily increased through the 20th century (based on comparison with aerial photos and mapped glacier extents (Kjeldsen et al., 2015) to the 2010s, the rate of increase has largely stabilized at a high level in the last few years. However, the spatial pattern of discharge varies through time and space. Initial high discharge numbers in the 2000s were driven by accelerations that later slowed in outlet glaciers in especially western Greenland but additional accelerations

in ice flow speeds at other outlets are sufficient to compensate and keep the overall discharge numbers high. Note that these figures do not include meltwater runoff from surface melt.

Taken together, the modelled runoff and ice discharge figures given in this section indicate that Greenland adds on average between 680 to 1000 Gt yr$^{-1}$ of fresh water to the oceans. However, the spatial variability in ice discharge and runoff complicates the interpretation of implications for the Arctic freshwater balance. The main regions of accelerating ice loss in Greenland drain out to the North Atlantic particularly in the high melt and high calving regions of western and south east Greenland. There has also been an observable increase in both calving and runoff from the outlet glaciers of northern Greenland (Hill et al., 2018; Solgaard et al., 2018; Shepherd et al., 2019, Extended Figure 4), which directly drains to the Arctic Ocean. Mankoff et al. (2019) estimate a stable ~26 Gt yr$^{-1}$ of ice discharge per year in the northern Greenland drainage basin that drains directly to the Arctic Ocean basin. This figure does not include surface melt and runoff but analysis by Fettweis et al. (2020) indicates about the same amount as an annual gain by SMB processes in the same basin up until 2013 but declining thereafter as surface melt has increased in this region.

Glaciers in the Canadian Arctic Archipelago draining into the same region as Northern Greenland have seen a succession of ice shelf collapses and associated changes in the fjords most likely related to sub-shelf melting and increased atmospheric air temperatures in the region since the 1950s (e.g., Copland et al, 2007, Gardner et al., 2011). Glaciers in Svalbard (e.g., Noel et al., 2020) and the high Russian Arctic have also shown consistent mass loss trends (e.g., Moholdt et al., 2012), indicating an increase in freshwater contribution from the smaller Arctic glaciers in the region directly into the Arctic Ocean basin but their contribution is an order of magnitude smaller than from Greenland.

The analysis of Arctic freshwater flux from land ice presented by Bamber et al. (2018) reaches a similar conclusion. The Bamber study estimates that, including land ice from other parts of the Arctic as well as the Greenland ice sheet, the total freshwater flux is around 1300 Gt yr$^{-1}$ in the period since 2010. They also identify a marked increase in runoff and discharge compared to a climatology period of 1960 to 1990. They also note that the distribution of the freshwater flux is not even around Greenland spatially, but also temporally, with both runoff and iceberg discharge peaking in summer but being rather low (though not zero) in winter. Compared to the other fluxes therefore, ice sheet and glacier discharge is a rather minor source of freshwater.

## 2.5    Ocean Transport Through Gateways

The latest reviews of the Arctic freshwater budget and fluxes (e.g., Haine et al., 2015; Carmack et al., 2016; Østerhus et al., 2019) conclude that observations of liquid freshwater transport through the Bering, Davis and Fram straits do not show significant trends between 1980-1990 and the 2000s. A recent study by Woodgate (2018) has shown that the Bering Strait exhibited a significant increase in volume and freshwater import to the Arctic between 2001 and 2014. Florindo-Lopez et al.

(2020) analyzed several decades of summertime hydrographic data at the eastern side of the Labrador Sea to find that freshwater transports in the boundary current were generally lower in the period mid-1990s to 2015 than the pre-1990s transports. The long-term variability was of the order of 30 milli-Sverdrup (one Sverdrup or Sv = $10^6$ $m^3$ $s^{-1}$).

Polyakov et al. (2020) have described the contrasting changes in the Eurasian and Amerasian basins, where the latter has shown increasing stratification in recent years. They relate this to an increased import of low-salinity waters through the Bering Strait (see Woodgate, 2018). In the Eurasian Basin, Polyakov et al. (2020) relate the weakening stratification and enhanced sea ice melt, a process referred to as the Atlantification of the Arctic (Polyakov et al., 2017), to injection of (warmer) relatively salty water from the Barents Sea into the Eurasian Basin halocline, flowing at shallower depths. Although they do not show any clear link to the Fram Strait imports, they find a small but statistically significant correlation between observed salinity in the Eastern Eurasian Basin halocline and the northern Barents Sea upper water column. These findings are consistent with the box model estimates of Tsubouchi et al. (2021), there appears to be no trend in volume fluxes at the boundaries, and no evidence for a dominant link between changes in the freshwater fluxes at the boundaries and changes in the upper Arctic Ocean, this is also true for the Atlantic water volume inflow.

## 3    Redistribution of Arctic Freshwater

The large-scale freshwater redistribution in the Arctic is mainly caused by the oceanic flows near the surface and in the upper ocean, up-to the lowermost extent of the Arctic halocline. It is governed by the two co-dependent and interactive components: wind-driven circulation and density-driven circulation. The wind distributes the fresh water through the advection in the Ekman layer, Ekman upwelling and downwelling and mixing. The ocean density gradients due to river runoff, precipitation and sea ice processes act through geostrophic density-driven flows, mixing of the ocean interior by lateral ocean eddies and through shelf topographic and tidal mixing and shelf cascading - all these processes impacting fresh water are discussed below.

### 3.1    Wind-driven circulation

The wind-driven circulation in the Arctic features: (i) the Beaufort Gyre (BG), a large-scale anticyclonic (clockwise) ocean gyre which occupies the Beaufort Sea and the Canadian Basin of the Arctic Ocean at the most extent; (ii) the cyclonic (anticlockwise) circulation on the Atlantic side of the Arctic Ocean (Nansen and Amundsen basins) and (iii) cyclonic ocean flows in the Siberian Shelf Seas. The Trans-Polar Drift (TPD), a large-scale stream which constitutes of the oceanic and sea ice coherent flows, has its sources in the Siberian Shelf Seas and follows across the North Pole to the Fram Strait. TPD can be found from the surface to the depth of the upper intermediate waters and until recently assumed to be a slowly (order of years to decades) varying flow, although sea ice retreat may destabilize sea ice and oceanic flows in TPD (e.g., Belter, 2021;

Krumpen et al., 2019). The wind-driven circulation produces local accumulation or thinning of the surface layer (Timmermans and Marshall, 2020).

Although, the exchanges with the Atlantic and Pacific influence the large-scale salinity gradients across the Arctic Ocean (Polyakov et al., 2020), the combined effects of the density-driven and wind-driven circulations primarily drive a strong freshwater gradient through the Arctic, of up to 25 m freshwater equivalent (Rabe et al., 2011), with a maximum freshwater content in the Beaufort Gyre and a minimum in the Nansen Basin towards the Barents Sea. Morison et al. (2012) and Alkire et al. (2007) in particular have shown the regional changes in steric height, by driving near-surface geostrophic currents, and sea level pressure respectively, can redistribute relatively fresh water near the surface along the boundaries of the deep basin and the shelves. In addition, Morison et al. (2021) recently provide a longer-term perspective on freshwater distribution and stress the importance of the cyclonic mode of ocean circulation on the Atlantic-Eurasian side of the Arctic Ocean, in addition to the conventionally emphasized Beaufort Gyre. The cyclonic mode is characterized as the first empirical orthogonal function (EOF) of ocean surface height variability (dynamic heights, 1959-1989 and satellite DOT, 2004-2019). In the cyclonic phase, surface depression, reduced freshwater content, and cyclonic circulation occurred on the Atlantic-Eurasian side of the Arctic Ocean. The freshwater loss in this area offsets freshwater gains in the Beaufort Gyre with the entire Arctic basin. The ocean cyclonic mode is related to the positive polarity of the AO with order 1-year latency, and has become more prominent under the long-term changing tendency of winter AO toward a positive phase since 1990 (see Figure 4).

Recent studies suggest that the Beaufort Gyre has stabilized or reached a new normal high-freshwater content state. Dewey et al. (2018) attributes this to a switch from a system driven by surface ice- and wind-stress that affects a passive ocean, to one where it is the ocean that drives the ice (often in the absence of wind). Zhong et al. (2019) in contrast attribute it to higher energy input to the ocean, and suggest that the transition is not complete, i.e., the Beaufort Gyre is not "saturated" yet. Zhong et al. (2019) further concludes that the recent increase in cyclonic activity reduces this energy input, and hence should result in future decrease of freshwater stored in the Beaufort Gyre. This surface circulation transports meteoric water (and hence nutrients, e.g., Bluhm et al., 2015) throughout the Arctic. On average, 10% of the Arctic surface waters are made up of meteoric waters (shallower than ~200 m depth; see Forryan et al. 2019, their figure 5b) and this number has so far been constant since the early 2000s (Alkire et al., 2017; Proshutinsky et al., 2019).

## 3.2    Competing processes in the Beaufort Gyre

The Beaufort Gyre is a retainer of liquid freshwater in the Arctic Ocean, governed by three factors: wind stress, the dynamic feedback between ice motion and upper ocean currents (ice-ocean governor) and lateral eddy fluxes (Doddridge et al., 2019). Observations and an idealized two-layer model study indicate that the "ice-ocean governor", controlling Ekman pumping, is five times more important than eddy dynamics in regulating the retention and release of freshwater from the Beaufort Gyre (Meneghello et al., 2020). Dewey et al. (2018) indicate Ekman pumping and the ice drag feedback mechanism stabilize ice

and ocean velocity at time scales of less than a week while eddy propagation feedback has time scales measured in years. Regan et al. (2020) have shown that the mean kinetic energy dominated over eddy kinetic energy (isopycnal slope / potential for baroclinic instability) in governing Beaufort Gyre dynamics during the spin-up in the past one to two decades. Armitage et al. (2020) predict that eddies will become more important in stabilizing the Beaufort Gyre. In addition, idealized simulations with and without a continental slope by Manucharyan and Isachsen (2019) demonstrate that eddy dynamics prevail in the Beaufort Gyre only in the presence of the slope. Further, Liang and Losch (2018) show not only that the positive feedback loop "enhanced vertical mixing = less sea ice" reduces halocline stratification and brings more salt to the deep

Arctic.

## 3.3    Ekman pumping and wind mixing

The wind also contributes to vertical redistribution via wind-driven coastal up and downwelling. On average, only the Laptev and Kara are dominated with downwelling; the rest of the Arctic, especially the Amerasian basin, is upwelling dominated (Williams and Carmack, 2015). However, bathymetric features can reverse the sign of this Ekman transport (Randelhoff and Sundfjord, 2018; Danielson et al., 2020). Even more relevant for freshwater, at locations where upwelling occurs, river plumes are pushed offshore (Williams and Carmarck, 2015; Våge et al., 2016). Downward flows of water can be generated by the wind or by an increase in density that destabilizes the water column. In addition, as mentioned in Section 2.3, strong storm forces Ekman upwelling, not only along the coast but also within the basin, which advect deeper layer warm and salty

water to the upper ocean layer (or in other words, cause a shoaling of the Pacific/Atlantic water layer) and, in turn, results in a salinization (Peng, et al., 2021). It is worth mentioning again that the storm-driven upwelling and increase in mixing also enhance sea ice melt, increasing upper mixed layer freshwater. The final vertical freshwater distribution depends on the competition between these two processes.

The wind also impacts the depth of the mixed layer. The Arctic surface mixed layer varies both seasonally and geographically, as reviewed by Peralta-Ferriz and Woodgate (2015). Using all available observations from 1979 to 2012, Peralta-Ferriz and Woodgate (2015) find a shoaling trend in the whole Arctic in winter; in summer the mixed layer trend is a deepening in ice free parts of the Barents and Beaufort Seas, but also a shoaling in the Eurasian basin. Polyakov et al. (2017) found an opposite trend using moorings and ice-tethered profilers: an increased winter convection caused by sea ice formation over a weakened stratification in the eastern Eurasian basin. They argue that the entire Eurasian basin is becoming

similar to the Atlantic sector of the Nansen basin and hence dubbed this phenomenon "the Atlantification of the Arctic" (or more recently, "the Borealisation of the Arctic").

## 3.4    Dense water cascading and tidal mixing

On the Arctic shelf, dense water can form as a result of cooling or brine rejection during sea ice formation, especially in polynyas (Ivanov et al., 2004). Cascading plumes entrain waters during their descent, explaining how cascading of cold and

saline surface waters can result in warmer (if entraining Atlantic water) or fresher (if entraining halocline) deeper levels (Backhaus et al., 1997). Preconditioning of the shelf waters due to the mixing with the upwelled Atlantic water also can result in the cold and saline cascading plumes (Luneva et al., 2020). Furthermore, cascading is becoming more common in the Arctic; it is more effective in mixing and ventilating upper and low intermediate Arctic waters than open ocean deep convection and can reach deep into the water column (e.g., Luneva et al., 2020). Cascading and entrainment in the Beaufort Sea during upwelling events re-injects cold and fresh water into the halocline (Ivanov et al., 2004). Janout et al. (2017) observed shelf processes and the modification of warm Atlantic Water leading to flux of the modified water, and hence an effective freshwater flux, toward the basin. From two expeditions in 2013 and 2014 and one year of mooring deployment in between, Janout et al. (2017) found a dual behavior in Vilkitsky Trough, between the Kara and Laptev Sea: strong winds can cause an upward diversion of the along-slope freshwater transport onto the shelf; the addition of sea ice formation results in the formation of water with a higher density than that found at 3000 m, suggesting possible sinking of these waters to the Nansen basin.

Tidal mixing of the waters on the Arctic continental shelves and over the steep shelf slopes topography has been shown to be of importance for the freshwater fluxes between the shelves and the ocean interior and for decadal variations and long-term decline of sea-ice cover (Luneva et al., 2015, 2020; Rippeth et al., 2015). The further sea ice retreat results in an increased upper-ocean shear and mixing, leading to enhanced ventilation of the Atlantic Water layer and impact on the sub-surface fresh water (Polyakov et al., 2020).

## 4    Summary

Our review of recent work suggests that Arctic freshwater content in the 2010s has stabilized relative to the 2000s. This stabilization is due in part to the compensation between an increase in the Beaufort Gyre and a decrease in the rest of the Arctic Ocean. However, large inter-model spread in the ocean reanalyses and uncertainty in the observations used in this study prevents a definitive estimate of the degree of this compensation. Inferred from the published literature, the most notable differences between the 2010s and the 2000s are:

1) the Arctic Oscillation and moisture transport into the Arctic are in-phase and have a positive trend;
2) the sea ice cover has transitioned to an increasingly seasonal and mobile, the impacts of this variability on the Arctic Ocean and atmosphere are still being debated;
3) mass loss from the Greenland ice sheet and other Arctic glaciers has increased, including in the northern region that drains directly into the Arctic Ocean;
4) vertical mixing of Atlantic Water into the deep Arctic has increased in the eastern Eurasian Basin where imported warm Atlantic waters have shoaled and the halocline has weakened.

This review also suggests that large uncertainties remain in quantifying regional patterns, changes, and individual contributors to freshwater content variability, motivating the need for long term monitoring, in-situ in rivers and ocean, and from space. Increased sample size of freshwater content and budget observations with time would help distinguish climate change forced long-term trends from internal low frequency variability.

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

**Code availability**
All code used for the present study is freely available by request.

**Data availability**
References that detail how to access the ORA model output, satellite observations, and in-situ observations are cited in the manuscript.

**Author contributions**
All authors contributed to defining the scope of this project and the writing of the manuscript.

1040    **Competing interests**
The authors declare that they have no conflict of interest.