# Peer review of "Freshwater in the Arctic Ocean 2010-2019"

_Ocean Science, 2020_

## Referee Comment (RC1) · Anonymous Referee #1 · 30 Dec 2020

**Summary**

This manuscript presents a useful updated freshwater budget for the Arctic Ocean. All components of the Arctic freshwater budget are reviewed, and some aspects that are lacking in the literature, such as the vertical redistribution of freshwater, are discussed in an interesting manner. The discussion of the meaning of "ocean freshwater flux" in the introduction is particularly useful. The text is generally clear and comprehensive, however several of the main points that are made in the abstract, introduction, and summary are not effectively conveyed in the manuscript or shown in the figures that are presented. This study will be a useful contribution to the literature after a major revision that synthesizes results from the literature in a more consistent manner, and supports its conclusions in a clear fashion.

**Major comments**

[Figure]

The authors argue that the trend in Arctic freshwater content stabilized in the 2010s due to increased compensation between the Beaufort Gyre and the remaining Arctic. However, that is not immediately apparent in any of the figures. Only one of the reanalysis products presented in Figure 2 shows a compensating pattern in the freshwater anomaly, but it also does not reproduce the observed storage trends. The multi-model mean shows a small amount of compensation on the shelves, but it is not clear that this balances the large storage in the Beaufort Gyre. The fidelity of the reanalysis products, or what we can learn from them, is not really discussed in the text. From Figure 3 it appears that the freshwater content in the Beaufort Gyre stabilized in the 2010s, at the same time as the freshwater content of the full Arctic stabilized: no compensation is apparent in this figure.

The two time series presented in Figure 3 are from different data products and use different reference salinities. This is not justified or discussed in the text. Why not also show the Beaufort Gyre freshwater content from the hydrographic observations, as it is a subset of the full Arctic?

The definition of the study region varies throughout the manuscript, which makes it difficult to interpret the numbers that are presented. For example, at Line 88, the authors state "in this study, the reference salinity used is 34.8 psu and freshwater content is calculated over the area north of 70N". However, in the Figure 3 caption a different domain and reference salinities are used. At Line 188, the authors give the impression that the Nordic Seas should be included in the analysis, but it is unclear that they have done this.

Runoff from the Greenland Ice Sheet has been considered in previous assessments, such Haine et al. 2015, Proshutinsky et al. 2015, so it is not particularly compelling to say that this aspect has been overlooked previously. It is also not made clear how freshwater fluxes from Greenland enter the budget, since much of Greenland is to the south of the Arctic (Mediterranean).

In the abstract and the summary the authors state that the import of subpolar waters into the Arctic has increased, yet this is not discussed substantially in the manuscript.

In general, it would be useful to include summary sentences at the end of each section.

**Minor Comments**

L26: You could specify "Arctic freshwater content" rather than "Arctic freshwater" here.

L41: This first sentence could be split into at least two.

L55: There may be a formatting issue here, but it seems S is being used for both salinity and salinity anomaly.

L58: What do you mean by "verbal"?

L81: This clause is confusing, starting from "inverse imprint"

L89: Can you comment a little bit more on why you choose 34.8 and how to interpret freshwater content and fluxes in this framework? Another significant freshwater framework that is missing in this discussion is that presented in Wijfells et al. 1992.

L167: Not sure this is a fair comparison, as Morison et al. 2007 present a very different time span. What is meant by "complex variability?"

L173: Which "difference in annual cycle" are you referring to?

L211: It is a bit confusing to include river discharge in this section after discussing river discharge in the previous section. Please clarify the links between these sections and which trends may be consistent between them. Boisvert et al. 2015, 2018 could also be referenced in this section.

L239: Could you clarify how sea ice age is converted into volume and how to interpret these results?

L255: Spall 2019 "Dynamics and Thermodynamics of the Mean Transpolar Drift and Ice Thickness in the Arctic Ocean" may be a useful reference to include here.

L260: Typo, should read "on par"

L265: Might add "before reaching the Transpolar Drift" to the end of the sentence.

L268: Please clarify this sentence. Does the decrease in sea ice extent cause a delay of freeze up?

L272: Was the snow depth greater than the climatology? Please clarify this sentence.

L299: In this section , you could also discuss the trends in Atlantic Water entering the Arctic as shown in Tsubouchi et al. 2020, for example.

L315: What is the "but" referring to here? Please rewrite this long sentence.

L316: Isn't the freshwater gradient in the Arctic caused by the difference in salinity between the Pacific and the Atlantic? It seems odd to credit the circulation with this gradient. Please clarify.

L321: May want to swap word order to "yet unreached".

L334: What is meant by more effective? Please explain and/or provide a reference.

L337: Please expand and clarify the synthesis of the Janout et al. 2017 study.

---

## Referee Comment (RC2) · Anonymous Referee #2 · 10 Jan 2021

R E V I E W

of the manuscript (os-2020-113) entitled "Freshwater in the Arctic Ocean 2010-2019" by A. Solomon et al.

The manuscript major focus is on variability of Arctic freshwater in the 2010s. This topic is of high scientific importance. I have a few minor comments listed below. I recommend minor revision.

Comments: 1. My understanding is that the Beaufort Gyre is a part of the Amerasian Basin. The authors state otherwise (see abstract and ll 120-125). 2. Page 12, line 247. Should "Sea" start with lower case "s"? 3. I tend to disagree with the statement ll 309-311 (p. 14) that Polyakov et al (2020) did not provide evidence for the link between the Barents Sea imports and salinity (freshwater) change in the halocline of the eastern Eurasian Basin. See their Fig 12, upper panel which compares salinity of the eastern

[Figure]

Eurasian Basin with salinity from the northern Barents Sea (lagged, led by one year). The time series show statistically significant link. This, plus well established pattern of the ocean circulation, provides reasonable evidence to state that changes in the eastern Eurasian Basin halocline show a fingerprint of changes in the upstream areas (notably northern Barents Sea). I suggest the authors modify this statement. 4. P. 15, l. 323: Please define surface waters: this percentage depends heavily on the thickness of the layer. 5. Same page, line 325. I tend to disagree with the statement that the wind is largely responsible for the subsurface horizontal redistribution of freshwater. If subsurface is defined as halocline, the circulation in the halocline is more like circulation in the Atlantic water layer and not the surface mixed layer. It is more complex and has a strong impact of topography and density.
* * *

---

## Short Comment (SC1) · 11 Jan 2021

This paper addresses a parameter that the authors call "Fresh water . . . in the ocean". Instead of analysing observed salinity changes in the Arctic Ocean and concluding from these on possible causes like mixing with or advection of fresh or low salinity waters, the authors present a useless parameter, "freshwater in the ocean", and therefore the paper does not provide unique results.

In the framework of TEOS-10, freshwater (furtheron FW) content in the ocean is defined in a unique way as FW=1-SA with absolute salinity SA given in g/kg (IOC/SCOR/IAPSO 2010). That is, almost the entire mass of ocean water is freshwater. What the authors are presenting in this paper are, however, fractions of FW defined by a reference salinity which is an arbitrary parameter by definition.

They analyse time series of these FW content fractions. As comparisons of freshwater fractions in general, also the course of such FW fraction time series is arbitrary. The authors admit the arbitrariness of the FW fraction (line 57ff) and refer to Schauer and Losch (2019; US19 furtheron). US19 illustrate the severe consequences of this arbitrariness which, however, are ignored by Solomon et al.

Despite recognizing the arbitrariness of FW fraction terms and their comparisons, the authors apply the freshwater fraction concept because "oceanographers have long been accustomed to [its] use" (line 53) and for "traditional" reasons (lines 87-88). I don't consider "being accustomed" or "tradition" sound scientific categories - particularly not so, when scientific arguments speak against them.

Although Solomon et al. nowhere determine pure freshwater fluxes (volume/time) to the ocean, they claim in the introductary chapter that the determination of fluxes of pure freshwater requires a specific "scaled" salinity (lines 68-77). But also this claim is wrong. According to the Knudsen theorem fluxes of PURE freshwater into or out of the ocean, such as e.g. precipitation, can be determined independently of the choice of the reference sality (Sref). Anyway, Solomon et al. don't compute fluxes of pure freshwater but the content of FW fractions and their variability.

Determining FW fraction content variability for a fixed ocean water volume remains ambiguous in the size but not in the sign. E.g. statements like in line 111, the FW increase being 25%, could as well be 15% or 35%, depending on the choice of Sref (see Table 1 in US19).

In their figures Fig. 2 and 3, however, Solomon et al. consider ocean volumes that are not fixed since they regard the upper layer down to a certain isohaline. The depth of the halocline can obviously vary. Calculating the content over a variable volume means, that also the sign of variation might change with changing Sref (see Fig. 4 of SL19; the formulation of varying volume is analogue to that of FW fraction transports through variable ocean currents.).

The authors could avoid all the ambiguity and arbitrariness of FW fractions and their

temporal evolution if instead they showed integrated salinities over one or several fixed layers. They could discuss how the varying salinity relates to salt flux divergence (either due to vertical mixing or due to advective divergence) due to the different external pure FW or low (known) salinity water input. This discussion would be built on sound absolute quantities and would make this article a reliable resource for future reference.

References IOC/SCOR/IAPSO, 2010: The international thermodynamic equation of seawater – 2010: Calculation and use of thermodynamic properties. Intergovernmental Oceanographic Commission, Manuals and Guides 56, UNESCO, 196 pp.,http://www.teos-10.org/pubs/TEOS-10_Manual.pdf.

Schauer, U. and Losch, M.: "Freshwater" in the Ocean is Not a Useful Parameter in Climate Research. J. Phys. Oceanog., 49(9), 2309–2321. doi:10.1175/JPO-D-19-0102.1, 2019.

---

## Referee Comment (RC3) · Anonymous Referee #3 · 12 Jan 2021

General The authors compute freshwater content (FWC) for the whole Arctic Ocean from hydrography and for the Beaufort Gyre from altimetry and GRACE. They argue that Arctic Ocean freshwater content has stabilized since 2010. They review various processes that might explain the timing of the stabilization including for example runoff, precipitation, moisture flux, and sea ice changes, but they do not seem to make a firm conclusion as to causality. Overall the paper seems incomplete and lacking in thoroughness.

I found it hard to reconcile the time series of Beaufort Gyre and whole-basin FWC with prior studies, particularly over the major freshening event centered on 2005-2008. In Figure 3, the increases in FWC for the whole-basin and Beaufort Gyre are virtually the same. Nothing in the time series records for the reviewed processes is shown to explain this change. Other studies have found that the increase in FWC in the Beaufort Sea is offset by decrease in FWC in the rest of the Arctic Ocean to the extent

that the increase in whole-basin FWC averages much less that Beaufort Sea FWC increase. Why is there this departure from past results? The authors don't recognize this question, and the author's FWC results are given with insufficient background on the hydrographic data distribution and details on the remote sensing to tell us. Further, no comparisons of hydrography-derived FWC with satellite-derived FWC are done for identical regions and identical times. Such comparisons could rule out issues due to using the different data types. I can't recommend publication without a clearer and more thorough evaluation of the data approaches, a more complete review of past interpretation of the 2005-2008 freshening and a stronger set of conclusions.

Specific Issues line 55 -It seems like this should read more like S)S'/Sref) where S'=S-Sref Line 85 - The Tsubouchi result is interesting. One might think the problem using a fixed reference salinity is dwarfed by the problem of hydrographic data coverage. The authors don't explain their remote sensing approach, butt might be noteworthy if the remote sensing of FWC with ocean bottom pressure versus dynamic ocean topography is really a measure of steric pressure, which for the Arctic Ocean correlates well with FWC relative to a fixed Sref.

Line 119 - One might add: For example, Rabe et al. (2011) and Morison et al. (2012) found that from the early to late 2000s, the increase deep basin freshwater content in the BG was largely balanced by a decrease in the rest of the Arctic Ocean.

Lines 120-127 - The differences between the models is significant, especially in Fig. 2b. Is this caused by using S=34.8 as a lower bound for the integration. It seems like this would be a problem particularly if the distributions of Atlantic water on the shelves are different between the models. Also, although nearly all the models show a freshening in the BG, many show no changes at all in the large regions of decreasing freshwater content described by Rabe et al. (2011) and Morison et al. (2012).

Line 145 - Measuring freshwater content change from altimetry and GRACE ocean bottom pressure was earlier described and validated with repeat hydrographic measurements by Morison et al. (2012)

Lines 154-166 - Caption for Fig 3 says nothing about GRACE OBP. Was it used in Fig. 3 or not?

Figure 3 and discussion of same - The BG record is said to be from altimetry and the whole basin is from hydrography. That's understandable given the time spans, but unfortunate. In recent years particularly, hydrography from the BG is plentiful but in situ observations, are few in the Amundsen, Nansen, and Makarov basins where we expect freshwater content has declined as BG FWC has increased. It would be helpful to compare satellite derived and hydrography derived FWC for identical BG and whole-basin regions to test the methodologies. It would also be illuminating to do the remote sensing comparisons over the whole basin including the Russian shelves where prior results and the modeling results of Figure 2 suggest decreased FWC acts to balance Beaufort Gyre FWC increase.

Figure 3. As stated above, his result, specifically the correlation of the increase in BG and whole-basin FWC from about 2004 to 2008, does not agree with the findings of Rabe et al. (2011) and certainly Morison et al. (2012) who found that the increase in BG freshwater over that time was largely compensated by decreasing trends in FWC in the Nansen, Amundsen, and Makarov basins, so much so that the whole deep basin average FWC trend could be accounted for by the loss in resident sea ice. In Fig. 3, the whole-basin FWC change around ∼2007-08 seems biased by the relative lack of observations outside the Beaufort Gyre to look like the BG FWC change. The result would be more convincing if we were given information on spatial sampling and possible sampling biases. Also the same-area technique comparisons mentioned above might make the result more credible.

Line 275 to 277. How does melt from Greenland get into the deep Arctic Basin? Virtually all the flow around Greenland is nominally southward, away from the Arctic Ocean.

Line 278 - IMBIE?

Lines 320-322 - Eddy fluxes have little to do with stabilizing the BG. Rather the feedback driven by the difference between surface geostrophic velocity and ice velocity balanced against dissipation by internal ice stress stabilizes the gyre at a time scale that is a small fraction of that due to eddy fluxes. See: Dewey, S., et al. (2018). Arctic ice-ocean coupling and gyre equilibration observed with remote sensing. Geophysical Research Letters, 45. https://doi.org/10.1002/2017GL076229

Lines 325-327 – The authors had better define what they mean by upwelling. Some, as seemingly here, mean upwelling in terms of what happens at the coast or surface (anticyclonic means upwelling) and some define it by what happens to the pycnocline in the center of the gyre (anticyclonic = downwelling).

Summary Section - I can agree that FWC content may have stabilized after 2010, but for reasons mentioned earlier, I don't think the changes in FWC 2005-2008 for the whole basin and the BG should be so similar. I can believe the ice is more mobile in recent years, but I don't understand how Greenland melt has any effect on Arctic Ocean FWC. The paper does not cover regional variability very well.

---

## Author Comment (AC1) · 17 Mar 2021

We thank all three anonymous reviewers and U. Schauer for reading this paper so carefully and providing such insightful comments. We have worked very hard to address your concerns as thoroughly as possible in the revised paper. In order to estimate to what extent the change in FWC outside of the Beaufort Gyre compensates for changes within the Beaufort Gyre in the 2010s, additional calculations were done with the ORAs to show FWC in the Beaufort Gyre and the rest of the Arctic, and with the satellite measurements below 82.5N. More details are provided about the data sources, and the satellite and in-situ FWC estimates are calculated more consistently. A spatial map of FWC from satellites is now included to compare the observational estimates more directly with the ORAs.

Our responses are shown with blue font below each comment.

**Anonymous Referee #1**

**Major comments**

1) The authors argue that the trend in Arctic freshwater content stabilized in the 2010s due to increased compensation between the Beaufort Gyre and the remaining Arctic. However, that is not immediately apparent in any of the figures. Only one of the reanalysis products presented in Figure 2 shows a compensating pattern in the freshwater anomaly, but it also does not reproduce the observed storage trends. The multi-model mean shows a small amount of compensation on the shelves, but it is not clear that this balances the large storage in the Beaufort Gyre. The fidelity of the reanalysis products, or what we can learn from them, is not really discussed in the text. From Figure 3 it appears that the freshwater content in the Beaufort Gyre stabilized in the 2010s, at the same time as the freshwater content of the full Arctic stabilized: no compensation is apparent in this figure.
It is correct to say that the reanalyses disagree on the extent and magnitude of trends outside the Beaufort Gyre. We have altered the maps in Figure 2 and now it is clearer that most of the products detect negative FWC trends along the Eurasian coastal seas. The magnitude of this trend is weak compared to the Beaufort freshening but nonetheless there is evidence of a slight compensation. Given that there is disagreement between models (Figure 2 time series of Beaufort and non-Beaufort FWC, for individual products), we have modified the text to express caution about making a definitive conclusion about the compensation.

2) The two time series presented in Figure 3 are from different data products and use different reference salinities. This is not justified or discussed in the text. Why not also show the Beaufort Gyre freshwater content from the hydrographic observations, as it is a subset of the full Arctic?
We address this issue by including the Arctic Ocean below 82.5N from satellites and use the same reference salinity of 35 for both satellite measurements and in-situ observations.

3) The definition of the study region varies throughout the manuscript, which makes it difficult to interpret the numbers that are presented. For example, at Line 88, the authors state "in this study, the reference salinity used is 34.8 psu and freshwater content is calculated over the area north of 70N". However, in the Figure 3 caption a different domain and reference salinities are used. At Line 188, the authors give the impression that the Nordic Seas should be included in the analysis, but it is unclear that they have done this.
These lines have been removed to prevent confusion about the regions and techniques used in this study.

4) Runoff from the Greenland Ice Sheet has been considered in previous assessments, such Haine et al. 2015, Proshutinsky et al. 2015, so it is not particularly compelling to say that

this aspect has been overlooked previously. It is also not made clear how freshwater fluxes from Greenland enter the budget, since much of Greenland is to the south of the Arctic (Mediterranean).

It is true that the majority of the runoff from Greenland flows south, however, more than 40% of the Greenland ice sheet drains north of 79 degrees (Hill et al., 2018) where both solid and liquid runoff are expected to impact the Arctic Ocean basin more directly and we therefore consider it worthwhile to include and highlight in the paper. We therefore also update the work of previous studies to look at the change in the recent decade.

Ref: Hill, E. A., Carr, J. R., Stokes, C. R., and Gudmundsson, G. H.: Dynamic changes in outlet glaciers in northern Greenland from 1948 to 2015, The Cryosphere, 12, 3243–3263, https://doi.org/10.5194/tc-12-3243-2018, 2018.

5) In the abstract and the summary the authors state that the import of subpolar waters into the Arctic has increased, yet this is not discussed substantially in the manuscript.

The near-surface freshwater import from lower latitudes has increased through the Bering Strait, as stated in Section 2.5: "A recent study by Woodgate (2018) has shown that the Bering Strait exhibited a significant increase in volume and freshwater import to the Arctic between 2001 and 2014."

In regard to the Atlantic sector, Polyakov (2017) says that Atlantic Water layer has shoaled and is warmer, most likely because it is warming at its source. But true, not that more has been / is being imported into the Arctic. This has been corrected in the summary and abstract. Also, from L308 onwards we add, "In the Eurasian Basin, they relate the weakening stratification and enhanced sea ice melt, a process referred to as the Atlantification of the Arctic (Polyakov et al., 2017), to injection of (warmer) relatively salty water from the Barents Sea into the Eurasian Basin halocline, flowing at shallower depths. Although they do not show any clear link to the Fram Strait imports, they find a small but statistically significant correlation between observed salinity in the Eastern Eurasian Basin halocline and the northern Barents Sea upper water column. Thus, in agreement with the box model estimates of Tsubouchi et al. (2020), there appears to be no trend in volume fluxes at the boundaries, and no evidence for a dominant link between changes in the freshwater fluxes at the boundaries and changes in the upper Arctic Ocean."

New reference:

Tsubouchi, Takamasa, Kjetil Våge, Bogi Hansen, Karin Margretha H. Larsen, Svein Østerhus, Clare Johnson, Steingrímur Jónsson, and Héðinn Valdimarsson. "Increased ocean heat transport into the Nordic Seas and Arctic Ocean over the period 1993–2016." *Nature Climate Change* 11, no. 1 (2021): 21-26.

6) In general, it would be useful to include summary sentences at the end of each section.

Thank you for this suggestion. We have included summary sentences when appropriate.

**Minor Comments**

1) L26: You could specify "Arctic freshwater content" rather than "Arctic freshwater" here.
Done.
2) L41: This first sentence could be split into at least two.
Thank you, we think the sentence reads well as is.
3) L55: There may be a formatting issue here, but it seems S is being used for both salinity and salinity anomaly.
Salinity anomaly is now denoted with "$\delta S$".
4) L58: What do you mean by "verbal"?
The sentence has been changed to read, "...reference salinity and values attributed to it are not rigorously mathematically and physically defined."
5) L81: This clause is confusing, starting from "inverse imprint"

The sentence has been changed to read, "Is "ocean freshwater flux" purely a mirage, therefore? Forryan et al. (2019) pursue the surface freshwater flux approach, noting that (as is well known, e.g. Östlund and Hut, 1984) evaporation and freezing are distillation processes that leave behind a geochemical imprint via oxygen isotope anomalies on the affected freshwater in the sea ice and seawater. In the case of evaporation, distillation (here, isotopic fractionation) preferentially removes lighter oxygen isotopes from seawater, leaving behind in the seawater a proportion of heavier isotopes. The lighter isotopes that are now in the atmosphere return to the land or sea surface as precipitation. Those falling on land can (eventually) transfer from land to sea by river runoff or by other glacial processes, or by further cycles of evapo-transpiration and precipitation. For sea ice, the ice contains the lighter isotopes while heavier isotopes are contained in the brine that drains out of the ice during freezing, to re-enter the ocean. The isotopically-lighter meteoric fractions are used to quantify freshwater that originates from the atmosphere (directly or indirectly), and the isotopically-heavier fractions similarly quantify the signal of brine rejected from sea ice, and thereby the amount of ice formed from that seawater."

6) L89: Can you comment a little bit more on why you choose 34.8 and how to interpret freshwater content and fluxes in this framework? Another significant freshwater framework that is missing in this discussion is that presented in Wijfells et al. 1992.

The selection of the reference salinity (Sref=34.8) for the Beaufort Gyre region is in accordance with previous studies focusing on the freshwater content of the region (Proshutinsky et al., 2009). The same reference has been used in an earlier study to investigate the freshwater sinks and sources in the Arctic (Aagard and Carmack, 1989). The FWC anomalies estimated here is as a measure of the amount of liquid freshwater accumulated or lost from the water column column bounded by the 34.8 isohaline at depths. Note that there are other approaches used which are independent of a reference salinity, for example in Wijfells et al. (1992) where they focused on the global distribution of freshwater transport in the ocean based on the integration point (reference point) in the Bering Strait. It should be noted that all of these methods and the use of different reference salinities (for example Dickson et al., used Sref=35.2 to study inflowing Atlantic Water) have merits and limitations based on the choice of intent and application (Carmark et al., 2008).

Carmack, E., F. McLaughlin, M. Yamamoto-Kawai, M. Itoh, K. Shimada, R. Krishfield, and A. Proshutinsky (2008), Freshwater storage in the Northern Ocean and the special role of the Beaufort Gyre, in Arctic- Subarctic Ocean Fluxes: Defining the Role of the Northern Seas in Cli- mate, edited by R. R. Dickson, J. Meincke, and P. Rhines, pp. 145–170, Springer, New York.

Wijffels SE, Schmitt RW, Bryden HL, Stigebrandt A (1992) Transport of freshwater by the oceans, Journal of Physical Oceanography, 22, 155–162.

7) L167: Not sure this is a fair comparison, as Morison et al. 2007 present a very different time span. What is meant by "complex variability?"

This refers to the interannual variability seen after 2007. The sentence is changed to "complex interannual variability".

8) L173: Which "difference in annual cycle" are you referring to?

This sentence has been removed.

9) L211: It is a bit confusing to include river discharge in this section after discussing river discharge in the previous section. Please clarify the links between these sections and which trends may be consistent between them. Boisvert et al. 2015, 2018 could also be referenced in this section.

We think the reference to river runoff in Section 2.2 is required to show the relationship between the different freshwater sources. A reference to Boisvert et al. 2018 has been added.

10) L239: Could you clarify how sea ice age is converted into volume and how to interpret these results?

Sea-ice age can be related to thickness. Given the sea-ice concentration, you can estimate volume. Liu et al. (2020) states: "The relationship between ice age and ice thickness is first established for every month based on collocated ice age and ice thickness from submarine sonar data (1984–2000) and ICESat (2003–2008) and an empirical ice growth model."

Further, they use a satellite-based product providing sea-ice age, and "The ice age category represents how long in years the sea ice has existed since its first appearance, which is estimated through Lagrangian tracking of the ice from week to week using gridded ice motion vectors (Maslanik et al., 2007, 2011; Tschudi et al., 2019b)." Further details can be found in Liu et al.

11) L255: Spall 2019 "Dynamics and Thermodynamics of the Mean Transpolar Drift and Ice Thickness in the Arctic Ocean" may be a useful reference to include here.

Spall reference added.

12) L260: Typo, should read "on par"

Spelling corrected.

13) L265: Might add "before reaching the Transpolar Drift" to the end of the sentence.

The sentence has been modified as suggested.

14) L268: Please clarify this sentence. Does the decrease in sea ice extent cause a delay of freeze up?

Not necessarily. We modified the sentence as "Although the delay of freeze up during early winter, partly depending on the anomalies of oceanic and atmospheric circulations (e.g., Kodaira et al., 2020), would cause a delay of snow accumulation on sea ice, …"

Reference:

Kodaira, T., Waseda, T., Nose, T., and Inoue, J.: Record high Pacific Arctic seawater temperatures and delayed sea ice advance in response to episodic atmospheric blocking. Sci. Rep., 10, 20830, https://doi.org/10.1038/s41598-020-77488-y, 2020.

15) L272: Was the snow depth greater than the climatology? Please clarify this sentence.

The sentence was modified as: "In the Atlantic sector, precipitation associated with six major storm events in 2014/2015 during the N-ICE2015 field campaign (Merkouriadi et al., 2017) caused the snow depth to be substantially greater than climatology."

16) L299: In this section, you could also discuss the trends in Atlantic Water entering the Arctic as shown in Tsubouchi et al. 2020, for example.

From Tsubouchi et al. 2021 AW volume inflow does not have any significant trend. We already discussed the trends in Atlantic Water when referring to Polyakov et al. (2020) L306, but the reviewer is right that more should be written and references are missing. We rephrased from L308 onwards: In the Eurasian Basin, they relate the weakening stratification and enhanced sea ice melt, a process referred to as the Atlantification of the Arctic (Polyakov et al., 2017), to injection of (warmer) relatively salty water from the Barents Sea into the Eurasian Basin halocline, flowing at shallower depths. They do not show any clear link to the Barents or Fram Strait imports. Thus, in agreement with the box model estimates of Tsubouchi et al. (2020), there appears to be no trend in volume fluxes at the boundaries, and no evidence for a dominant link between changes in the freshwater fluxes at the boundaries and changes in the upper Arctic Ocean.

New reference:

[Polyakov 2017 is already cited elsewhere in our manuscript]

Tsubouchi, Takamasa, Kjetil Våge, Bogi Hansen, Karin Margretha, H. Larsen, Svein Østerhus, Clare Johnson, Steingrímur Jónsson, and Héðinn Valdimarsson. "Increased

ocean heat transport into the Nordic Seas and Arctic Ocean over the period 1993–2016."
Nature Climate Change 11, no. 1 (2021): 21-26.

17) L315: What is the "but" referring to here? Please rewrite this long sentence.
The sentence has been rewritten to read,"...consisting mostly of the anticyclonic/convergent Beaufort Gyre and the cyclonic/divergent Transpolar Drift. The wind-driven circulation produces local accumulation or thinning of the surface layer..."

18) L316: Isn't the freshwater gradient in the Arctic caused by the difference in salinity between the Pacific and the Atlantic? It seems odd to credit the circulation with this gradient. Please clarify.
Thank you for pointing out that this is unclear. In fact, there are different sources for fresh water in the Arctic, relative to the most saline input, from the Atlantic / Nordic Seas through the Fram Strait and the Barents Sea: continental runoff, precipitation and Pacific Water through the Bering Strait. Pacific Water has slightly lower salinity than Atlantic Water, whereas the other sources are pure fresh water. The different sources are described in detail in Section 2. The Arctic circulation redistributes that relatively fresh water in solid and liquid form on the large scale via the Transpolar Drift and the Beaufort Gyre. The cross-Arctic gradient from low fresh water north of the Fram Strait and the Barents Sea to high fresh water in the Canada Basin stems from all of these having a lower salinity than the saline inflow of modified Atlantic Water and being redistributed by circulation. This has been illustrated in various publications cited in our manuscript, e.g., in Section 3. Both, the sources/sinks of salinity (i.e., Pacific and Atlantic exchanges, along with the continental runoff, sea ice melt/formation and precipitation less evaporation fluxes) play a role in setting up the cross-Arctic freshwater gradients (Haine et al., 2015). However, the interior Arctic fresh water re-distribution is mostly controlled by the ocean large circulations and, specifically, by convergence of the fresh water in the Beaufort Gyre and corresponding fresh water divergence away from the Siberian shelves. We made the following clarification in the text: "Although, the exchanges with the Atlantic and Pacific influence the large scale salinity gradients across the Arctic Ocean (Polyakov et al., 2020), the combined effects of the density-driven and wind-driven circulations primarily drive a strong freshwater gradient through the Arctic, of up to 25 m freshwater equivalent (Rabe et al., 2011), with a maximum freshwater content in the Beaufort Gyre and a minimum in the Nansen Basin towards the Barents Sea."

19) L321: May want to swap word order to "yet unreached".
The sentence has been modified as suggested.

20) L334: What is meant by more effective? Please explain and/or provide a reference.
We have explained the statement and added more details and the reference:
Preconditioning of the shelf waters due to the mixing with the upwelled Atlantic water also can result in the cold and saline cascading plumes (Luneva et al., 2020). Furthermore, cascading is becoming more common in the Arctic; it is more effective in mixing and ventilating upper and low intermediate Arctic waters than open ocean deep convection and can reach deep into the water column (e.g., Luneva et al., 2020).
The sentence has been rephrased to clarify: "Furthermore, cascading is more effective in the highly stratified Arctic Ocean than open ocean deep convection in reaching deep into the water column."

21) L337: Please expand and clarify the synthesis of the Janout et al. 2017 study.
The sentence has been changed to read, "From two expeditions in 2013 and 2014 and one year of mooring deployment in between, Janout et al. (2017) found a dual behaviour in Vilkitsky Trough, between the Kara and Laptev Sea: strong winds can cause an upward diversion of the along-slope freshwater transport onto the shelf; the addition of sea ice formation results in the formation of water with a higher density than that found at 3000 m, suggesting possible sinking of these waters to the Nansen basin."

**Anonymous Referee #2**

**Comments**:

1) My understanding is that the Beaufort Gyre is a part of the Amerasian Basin. The authors state otherwise (see abstract and ll 120-125).
   Thank you for pointing this out. The abstract has been changed to read, "due to an increased compensation between a freshening of the Beaufort Gyre and a reduction in freshwater in the rest of the Arctic Ocean" and lines 120-125 have been changed to read, "However, this freshening is partly compensated by a reduction in freshwater in the rest of the Arctic Ocean".

2) Page 12, line 247. Should "Sea" start with lower case "s"?
   Thank you but the spelling is correct.

3) I tend to disagree with the statement ll 309- 311 (p. 14) that Polyakov et al (2020) did not provide evidence for the link between the Barents Sea imports and salinity (freshwater) change in the halocline of the eastern Eurasian Basin. See their Fig 12, upper panel which compares salinity of the eastern Eurasian Basin with salinity from the northern Barents Sea (lagged, led by one year). The time series show statistically significant link. This, plus well-established pattern of the ocean circulation, provides reasonable evidence to state that changes in the eastern Eurasian Basin halocline show a fingerprint of changes in the upstream areas (notably northern Barents Sea). I suggest the authors modify this statement.
   Polyakov et al. (2020) did show the timeseries of the salinity in the northern Barents Sea and in the Eastern Eurasian Basin (their Fig 12). They did show a correlation coefficient of 0.41 for the Barents Sea lagging by 1 yr and state that this correlation is significant (stating the way this was determined in their methods). They did not, however, show any relation (or lack thereof) to the Fram Strait inflow. We adjusted the text to reflect the fact: "Although they do not show any clear link to the Fram Strait imports, they find a small but statistically significant correlation between observed salinity in the Eastern Eurasian Basin halocline and the northern Barents Sea upper water column."

4) P. 15, l. 323: Please define surface waters: this percentage depends heavily on the thickness of the layer.
   This sentence has been changed to read, "On average, 10% of the Arctic surface waters are made up of meteoric waters (shallower than ~200 m depth;  see Forryan et al. 2019, their figure 5b) and this number . . . ".

5) Same page, line 325. I tend to disagree with the statement that the wind is largely responsible for the subsurface horizontal redistribution of freshwater. If subsurface is defined as halocline, the circulation in the halocline is more like circulation in the Atlantic water layer and not the surface mixed layer. It is more complex and has a strong impact of topography and density.
   The role of the wind and other components of the climate system is extensively discussed in this section. To avoid repetitions, we rephrased:
   "The wind also contributes to vertical redistribution via wind-driven up and downwelling."

**Anonymous Referee #3**

**General  Comments:**

1) Overall the paper seems incomplete and lacking in thoroughness.
   Thank you for this comment. We have worked very hard to include a more thorough analysis of Arctic Ocean FWC for the 2010s relative to the 2000s using ORAs, previous studies, in-situ and satellite measurements.
2) I found it hard to reconcile the time series of Beaufort Gyre and whole-basin FWC with prior studies, particularly over the major freshening event centered on 2005-2008. In Figure 3, the increases in FWC for the whole-basin and Beaufort Gyre are virtually the same. Nothing in the time series records for the reviewed processes is shown to explain this change. Other studies have found that the increase in FWC in the Beaufort Sea is offset by decrease in FWC in the rest of the Arctic Ocean to the extent that the increase in whole-basin FWC averages much less that Beaufort Sea FWC increase. Why is there this departure from past results? The authors don't recognize this question, and the author's FWC results are given with insufficient background on the hydrographic data distribution and details on the remote sensing to tell us.
   The timeseries of FWC for the whole basin to the 34 isohaline is extended from Rabe et al. (2014). Details of the mapping procedure and the distribution of hydrographic stations is given there until 2012. Further data is based on the data sources listed in Table 2 of the revised manuscript. We have added the table and a reference to Table 2 in the caption of Figure 3. The timeseries will be updated in the future by adding further data sources and published elsewhere.
   The two timeseries in Figure 3 do not show the same trend from 2002 to 2015, indicating that the remainder is contributed by the rest of the basin outside the Beaufort Gyre (BG). It is further clear that the Beaufort Gyre dominates the increase in FWC, even though a significant increase is seen also outside the BG.
   References:
   Rabe, Benjamin; Schauer, Ursula; Ober, Sven; Horn, Myriel; Hoppmann, Mario; Korhonen, Meri; Pisarev, Sergey; Hampe, Hendrik; Villacieros, Nicolas; Savy, Jean Philippe; Wisotzki, Andreas (2016): Physical oceanography during POLARSTERN cruise PS94 (ARK-XXIX/3). Alfred Wegener Institute, Helmholtz Centre for Polar and Marine Research, Bremerhaven, PANGAEA, https://doi.org/10.1594/PANGAEA.859558
   Roloff, Albrecht; Rabe, Benjamin; Kikuchi, Takashi; Wisotzki, Andreas (2015): Physical oceanography from 49 XCTD stations during POLARSTERN cruise PS87 (ARK-XXVIII/4). Alfred Wegener Institute, Helmholtz Centre for Polar and Marine Research, Bremerhaven, PANGAEA, https://doi.org/10.1594/PANGAEA.853770
   Vogt, Martin; Rabe, Benjamin; Kikuchi, Takashi; Wisotzki, Andreas (2015): Physical oceanography from 15 XCTD stations during POLARSTERN cruise PS86 (ARK-XXVIII/3 AURORA). Alfred Wegener Institute, Helmholtz Centre for Polar and Marine Research, Bremerhaven, PANGAEA, https://doi.org/10.1594/PANGAEA.853768
3) Further, no comparisons of hydrography-derived FWC with satellite-derived FWC are done for identical regions and identical times. Such comparisons could rule out issues due to using the different data types.
   We address this issue by including the Arctic Ocean below 82.5N from satellites and use the same reference salinity of 35 psu for both satellite measurements and in-situ observations. In this way we can compare the satellite measurements with the in-situ observations and ORAs.
4) I can't recommend publication without a clearer and more thorough evaluation of the data approaches, a more complete review of past interpretation of the 2005-2008 freshening and a stronger set of conclusions.

The revised paper now includes a more thorough evaluation of the different data approaches by calculating FWC from satellites using the same reference salinity as the in-situ estimate, as well as, by including a spatial map of (2010-2017) minus (2002-2010) FWC from satellites to compare directly with the ORA results. A more detailed discussion of freshwater from Greenland Ice Sheet discharge is now included in Section 2.4. The conclusions are now based on a more thorough evaluation of the ORAs, previous studies, and new estimates of FWC from in-situ and satellite measurements.

**Specific Issues**

1) Line 55 -It seems like this should read more like S)S'/Sref) where S'=S- Sref Line 85 - The Tsubouchi result is interesting. One might think the problem using a fixed reference salinity is dwarfed by the problem of hydrographic data coverage. The authors don't explain their remote sensing approach, but might be noteworthy if the remote sensing of FWC with ocean bottom pressure versus dynamic ocean topography is really a measure of steric pressure, which for the Arctic Ocean correlates well with FWC relative to a fixed Sref.

   Salinity anomaly is now denoted with "$\delta S$". We agree with the reviewer that the problem of using a fixed reference salinity is dwarfed by the lack of coverage in the hydrographic observations. We have included a detailed discussion of the remote sensing approach in Section 1.1.

2) Line 119 - One might add: For example, Rabe et al. (2011) and Morison et al. (2012) found that from the early to late 2000s, the increased deep basin freshwater content in the BG was largely balanced by a decrease in the rest of the Arctic Ocean.

   Thank you, this sentence has been added.

3) Lines 120-127 - The differences between the models is significant, especially in Fig. 2b. Is this caused by using S=34.8 as a lower bound for the integration. It seems like this would be a problem particularly if the distributions of Atlantic water on the shelves are different between the models.

   The reviewer is right to comment on the model disagreements outside of the Beaufort gyre. We agree that inconsistencies in the representation of the incoming Atlantic water *may* explain why there is little agreement on the Eurasian Basin freshwater changes yet strong agreement in the Beaufort Gyre (where the Atlantic water has less influence). In response to this and other comments, as well as to further analysis, we can no longer conclude that the Beaufort gyre freshening is being compensated by FWC decreasing elsewhere in the Arctic. Future work will be necessary to understand the model differences.

   Also, although nearly all the models show a freshening in the BG, many show no changes at all in the large regions of decreasing freshwater content described by Rabe et al. (2011) and Morison et al. (2012).

   These two referenced studies both identify "large regions of decreasing freshwater content" in the Eurasian basin observations specifically for the latter half of the 2010s*. Given their target period is different and shorter than ours, a direct comparison is difficult. We also note the two datasets used in (Rabe et al, 2011) do not agree on the FWC trend in the Eurasian Basin, with observations finding weaker FWC decrease than the model used. Therefore, it is difficult to suggest there was previous agreement on FWC trends outside of the Beaufort Gyre.

   Moreover, these two studies do not show data for FWC trends for the shelf regions on the Eurasian side of the Arctic Ocean. We have altered the scale in Figures 2b&c, which now show decreasing freshwater content in these shelf regions in all but one model. This decreasing trend is relatively weaker than the freshening of the Beaufort Gyre but is a

feature which the models agree mostly agree on. Morison et al. (2011) studied the trend in the latter half of the 2010s, using satellite and in-situ data and a reference salinity of 34.87. Rabe et al. (2011) meanwhile studied the latter half of the 2000s compared to the 1990s, using satellite and model data and a reference salinity of 34.

4) Line 145 - Measuring freshwater content change from altimetry and GRACE ocean bottom pressure was earlier described and validated with repeat hydrographic measurements by Morison et al. (2012)
We thank the reviewer for pointing out the study by Morison et al., 2012. We have included this reference in the revised version of the manuscript. (Line 144-145). "The methodology which exploits the satellite derived ocean mass change and satellite altimeter data has been detailed in Gilles et al. (2012), Morison et al. (2012), and Armitage et al. (2016).

5) Lines 154-166 - Caption for Fig 3 says nothing about GRACE OBP. Was it used in Fig. 3 or not?
Yes. We thank the reviewer for correcting the mistake in the figure caption. "Anomalies of freshwater content (in $10^3$ km$^3$) in the Beaufort Gyre from satellite sea-surface height data analysis and **GRACE OBP data** (green) and in the whole Arctic Basin from objectively mapped in-situ hydrographic observations (blue).

6) Figure 3 and discussion of same - The BG record is said to be from altimetry and the whole basin is from hydrography. That's understandable given the time spans, but unfortunate. In recent years particularly, hydrography from the BG is plentiful but in situ observations, are few in the Amundsen, Nansen, and Makarov basins where we expect freshwater content has declined as BG FWC has increased. It would be helpful to compare satellite derived and hydrography derived FWC for identical BG and whole- basin regions to test the methodologies. It would also be illuminating to do the remote sensing comparisons over the whole basin including the Russian shelves where prior results and the modeling results of Figure 2 suggest decreased FWC acts to balance Beaufort Gyre FWC increase.
As explained in the text (Section 1.1), the altimeter data is not readily available in ice-covered areas. We have added FWC for altimeter data below 82N. FWC for the whole basin is not advisable due to the large uncertainties associated with the altimeter data in polar region, especially above the latitude band of 82N resulting in the polar gap (see Figure 1 in Raj et al., 2020).

7) Figure 3. As stated above, his result, specifically the correlation of the increase in BG and whole-basin FWC from about 2004 to 2008, does not agree with the findings of Rabe et al. (2011) and certainly Morison et al. (2012) who found that the increase in BG freshwater over that time was largely compensated by decreasing trends in FWC in the Nansen, Amundsen, and Makarov basins, so much so that the whole deep basin average FWC trend could be accounted for by the loss in resident sea ice. In Fig. 3, the whole-basin FWC change around ~2007-08 seems biased by the relative lack of observations outside the Beaufort Gyre to look like the BG FWC change. The result would be more convincing if we were given information on spatial sampling and possible sampling biases. Also the same-area technique comparisons mentioned above might make the result more credible.
Rabe et al. (2011) did not conclude that the FWC in different parts of the Arctic compensated around the middle of the 2010s. In fact, their Fig. 2 shows an increase in FCW inventories almost everywhere in the basin, between the time periods 1992-1999 and 2006-2008 (only JAS). Neither did they conclude that the increase in the FWC "could be accounted for by the loss in resident sea-ice". Sea-ice is potentially subject to enhanced melt but also decreased freeze-up and increased export. Rabe et al. (2011) only concluded that sea-ice melt and increased river water inflow partly caused the FWC changes, that were

dominated by decreasing average salinity in the Polar Mixed Layer and Upper Halocline, rather than a change in the thickness of that layer. Please see Haine et al. (2015) for a complete review of the potential relation between liquid and solid FWC. Multiple "sources" of liquid FWC increase are mentioned there, including changes in Precipitation-Evaporation, sea-ice melt and changes in the imports / exports through the various Arctic gateways. The bottom line is that it's not possible to identify the single most significant source due to errors in the source data. The data coverage during 2007 and 2008 was particularly good across the whole Arctic basin, due to various IPY-related ship expeditions and deployments of autonomous ice-tethered CTD profilers. Please see Rabe et al. (2014) and Rabe et al. (2011) for a discussion of mapping errors related to data coverage.

8) Line 275 to 277. How does melt from Greenland get into the deep Arctic Basin? Virtually all the flow around Greenland is nominally southward, away from the Arctic Ocean.

It is true that the majority of the runoff from Greenland flows south, however, more than 40% of the Greenland ice sheet drains north of 79 degrees (Hill et al., 2018) where both solid and liquid runoff are expected to impact the Arctic Ocean basin more directly and we therefore consider it worthwhile to include and highlight in the paper. We therefore also update the work of previous studies to look at the change in the recent decade. We have updated the Greenland ice sheet section to focus on the ice flux from this northern region in particular.

Ref: Hill, E. A., Carr, J. R., Stokes, C. R., and Gudmundsson, G. H.: Dynamic changes in outlet glaciers in northern Greenland from 1948 to 2015, The Cryosphere, 12, 3243–3263, https://doi.org/10.5194/tc-12-3243-2018, 2018.

9) Line 278 - IMBIE?

Added to line 278, "The Ice sheet Mass Balance Inter-comparison Exercise (IMBIE)...

10) Lines 320-322 - Eddy fluxes have little to do with stabilizing the BG. Rather the feed-back driven by the difference between surface geostrophic velocity and ice velocity balanced against dissipation by internal ice stress stabilizes the gyre at a time scale that is a small fraction of that due to eddy fluxes. See: Dewey, S., et al. (2018). Arctic ice-ocean coupling and gyre equilibration observed with remote sensing. Geophysical Research Letters, 45. https://doi.org/10.1002/2017GL076229.

We added the reference suggested by the reviewer and rephrased our reference to Zhong et al. (2019):

"Recent studies suggest that the Beaufort Gyre has stabilised or reached a new normal high-freshwater content state. Dewey et al. (2018) attributes this to a switch from a system driven by surface ice- and wind-stress that affects a passive ocean, to one where it is the ocean that drives the ice (often in the absence of wind). Zhong et al. (2019) in contrast attribute it to higher energy input to the ocean, and suggest that the transition is not complete, i.e. the Beaufort Gyre is not "saturated" yet. Zhong et al. (2019) further concludes that the recent increase in cyclonic activity reduces this energy input, and hence should result in future decrease of freshwater stored in the Beaufort Gyre."

11) Lines 325-327 – The authors had better define what they mean by upwelling. Some, as seemingly here, mean upwelling in terms of what happens at the coast or surface (anticyclonic means upwelling) and some define it by what happens to the pycnocline in the center of the gyre (anticyclonic = downwelling).

Note that the first sentence has been modified in response to a comment by reviewer 2. We here meant coastal upwelling, as the reviewer noticed. We added this precision:

"The wind also contributes to vertical redistribution via wind-driven coastal up and downwelling. On average, only the Laptev and Kara are dominated with downwelling; the rest of the Arctic, especially the Amerasian basin, is upwelling dominated (Williams and Carmack, 2015)."

**Summary Section**

1) I can agree that FWC content may have stabilized after 2010, but for reasons mentioned earlier, I don't think the changes in FWC 2005-2008 for the whole basin and the BG should be so similar.
   We have added a time series of satellite measurements in the Arctic Ocean below 82.5N, not including the Barents Sea because our methodology does not provide the best results in regions where the thermosteric component plays an important role in the steric sea level variability. Interestingly, these FWC estimates for the two regions produce similar FWC for 2005-2008. It is only after 2009 that these two time series diverge, indicating increased compensation after 2009.

2) I can believe the ice is more mobile in recent years, but I don't understand how Greenland melt has any effect on Arctic Ocean FWC.
   This is explained in detail in the response to Specific Comment #8 and in the revised paper Section 2.4. It is true that the majority of the runoff from Greenland flows south, however, more than 40% of the Greenland ice sheet drains north of 79 degrees (Hill et al., 2018) where both solid and liquid runoff are expected to impact the Arctic Ocean basin more directly and we therefore consider it worthwhile to include and highlight in the paper. We therefore also update the work of previous studies to look at the change in the recent decade. We have updated the Greenland ice sheet section to focus on the ice flux from this northern region in particular.

3) The paper does not cover regional variability very well.
   We address this issue by including satellite estimates of FWC in the Beaufort Gyre and the total Arctic below 82.5°N, and a spatial map of (2010-2017) minus (2002-2010) FWC from satellites.

**Response to comment posted by Ursula Schauer:**

We agree that it is necessary to use sound absolute quantities in scientific research and that there are ambiguities in using the freshwater fraction concept. However, we have already explicitly addressed SC1's point in our manuscript: we discuss the matter in the Introduction (LL 53-89), which includes reference to the reviewer's own 2019 publication. This is a review paper and most studies on this subject use the freshwater fraction concept, so we do the same in our manuscript when assessing these studies. Tesdal and Haine (JGR 2020, their section 2.3) confronted the same question and reached the same conclusion as us.

---

## Referee Report (RR1)

**Summary**

This manuscript presents a useful status update on the freshwater budget for the Arctic Ocean. All components of the Arctic freshwater budget are reviewed in a comprehensive and interesting manner. The authors made a significant effort to improve the analysis and presentation of results from its previous version, however, significant issues remain. This study will be a useful contribution after a major revision that presents the main results in a consistent and straightforward manner.

**Major comments**

The authors have softened their claim that there is fresh water compensation between the Beaufort Gyre and the rest of the Arctic in the abstract, however the presentation throughout the manuscript still needs significant improvement. Studies that suggest compensation seem to be preferentially cited (See L162 comment below) and some seem to be misrepresented (see L159 comment).

The analysis of the reanalysis products is much more comprehensively presented, but it would be very helpful to include boxes indicating which areas are being integrated over in addition to describing the regions in the Figure 2 caption. The same holds for Figure 3.

The satellite and in-situ fresh water estimates are very different during the period that this study focuses on, yet this is not discussed. The methods used for the satellite calculation are not presented (what empirical constants are used? What isohaline/reference salinity is being targeted?), so it is unclear how we might expect them to compare with insitu measurements. Compensation in the satellite estimate appears to decrease in the last part of the record. This is also not discussed. It is unclear why half of the Arctic is left out of (part of?) this analysis, and also unclear why an insitu estimate for the Beaufort Gyre is not presented. Presumably it is a subset of the full Arctic dataset? Please explain these analysis choices.

My recommendation is for major revisions because these observational analyses are a foundational part of this study and it is important that they be presented clearly, but note that the changes are much less significant than those recommended in the last revision.

**Specific Comments**
*Note: line numbers refer to the tracked-changes version of the manuscript.*

L67: Generally deltaS=S_ref-S, please check and revise.

L131: Only Greenland ice sheet contributions are discussed, this could be a good place to also introduce GIS runoff contributions.

L139: Not sure that adding the depths of the halocline and Atlantic layer is useful, as they vary substantially across the Arctic.

L159: Which studies are you referring to? (Proshutinsky et al., 2009; McPhee et al., 2009; Rabe et al., 2011; Haine et al., 2015)? It is inaccurate to state that these studies do not take redistribution or the Greenland Ice Sheet into account.

L161: Remove "(among other processes)" or explain which processes you are referring to.

L162: Rabe et al. 2011 and Morison et al. 2012 do not quantify the degree of compensation between the Beaufort Gyre and the rest of the Arctic Ocean. Wang et al. 2019 "Recent Sea Ice Decline Did Not Significantly Increase the Total Liquid Freshwater Content of the Arctic Ocean", show an updated time series from Rabe, which suggests that there has not been compensation between the Amerasian and Eurasian basins.

Figure 2: It would be helpful to show which areas are summed over to produce panel D in panel C.

L226: It would be useful to provide a brief review of the method used in Giles et al. 2012 and Armitage et al. 2016, and now in this paper, including the empirical constant used, for clarity.

L265: Why aren't these areas included in Figure 3? Please explain this further.

L278: Does this mean that the in-situ estimates at the end of the record may be underestimates? In this case, the discrepancy between the satellites and in-situ estimates may be even larger than what is presented here.

Figure 3: Please indicate where your Beaufort Gyre region in panel B. In the caption you state that the time series are calculated to the 34 isohaline using a reference salinity of 35. This is likely not the case for the satellite calculation, which uses an empirical constant to relate steric changes to freshwater changes and should be clarified. Again, why is the in-situ BG time series not shown?

L329: Please justify this statement and include a citation: "Precipitation over the Arctic is the main source of freshwater into the Arctic Ocean", including that from river discharge from the large continental drainage basins"

L456: Could reconsider the use of the word "Discharge" here as it is usually associated with only the solid component and you also discuss liquid runoff.

L497: It would be useful if you could explain and lay out why the total flux is larger than the net mass loss in a simpler fashion.

L509: Does Mankoff consider only solid freshwater flux? Or total? Please clarify.

L544: Translate Gt to mSv to make comparable with other sources.

L559: Could remove "as we do" since you are actively noting it in that sentence.

L628: Odd phrasing. Steric height changes do not redistribute fresh water, rather they are reflective of freshwater redistribution.

L640: Sentence is repeated.

L696: I didn't follow the last clause of this sentence, please clarify.

L706: Suggest weakening "is due" to "is potentially due", or "appears to be".

L714: Unclear why "to distinguish trends from low frequency variability" is included here.

---

## Author Response (AR2)

We thank the Editor, the two anonymous reviewers, and Jamie Morison for reading this paper so carefully and providing such insightful comments. We have worked very hard to address your concerns as thoroughly as possible in the revised paper. All estimates of FWC from ORAs, satellites, and in-situ observations have limitations. Our strategy has been to include all of these sources to provide a qualitative understanding of Arctic Ocean FWC in the 2010s. More details are provided about the in-situ data sources. The satellite, in-situ, and ORA FWC estimates are calculated more consistently. We appreciate the concern about not including the Barents and Nordic Seas in the satellite observations but the motivation for this is explained clearly in the responses below. A spatial map of the location of in-situ measurements is now included to compare the in-situ observations more directly with the ORAs and satellite observations. We have reorganized the section on redistribution.

Our responses are shown with blue font below each comment.

Comments from the Editor:

Line numbers from manus version 3)
1. In general there is a mix of selected isohaline for vertical integration (34.8 vs. 34) and units of FWC (km^3 vs m). Could you be more consistent in a review (perhaps reproduce figures consistently)?
   Thank you for this comment. All analyses presented in this paper now use the 34 isohaline for vertical integration. We have recalculated FWC in units of meters.
2. Li 35: do we have enhanced mixing in the ocean? Does Timmermans and Marshall 2020 cover this? I think this claim is overreaching. Perhaps supported in the eastern EB (Fig 11 of Polyakov et al. JClim 2020).
   Reference changed and sentence modified to indicate enhancement in mixing regionally.
3. In line 40 you could cite Carmack et al. BAMS paper.
   Reference added.
4. Li 43-44: it is a bold statement that ocean stratification determines sea ice growth.
   The word "determines" has been changed to "affects".
5. Li 45: I cannot follow how the part after ";" connects to the previous sentence. Perhaps a separate sentence could help.
   The sentence has been changed to read, "Freshwater in the Arctic Ocean plays a critical role in the global climate system; by impacting large-scale overturning ocean circulations (Sévellec et al., (2017), see Figure 1 showing basins and upper circulation); by changing ocean stratification that affects sea ice growth, biological primary productivity (Ardyna and Arrigo, 2020; Lewis et al., 2020),  and ocean mixing (Aagaard and Carmack, 1989); and by the emergence of freshwater regimes that couple variability in land, atmosphere, and ocean systems (e.g., Jeffries et al., 2013; Wood et al., 2013)."
6. Li 46-51: AO freshwater balance somewhat ignored the vertical mixing.
   Line 51 has been changed to read, "redistribution between Arctic basins and vertical mixing (e.g., Timmermans et al., 2011; Morison et al., 2012; Proshutinsky et al., 2015)."
7. Li 55-95: Discussion around freshwater. I am disappointed that you join many others and follow the flawed traditional approach instead of taking a step in the right direction in a review in 2021. Fine. But as you end this section (page 3) please also mention what the alternative is (e.g., ocean salinity and salinity flux divergence analyses).

We have added this sentence to clarify our approach, "This atmosphere-ocean surface freshwater flux is a key element of the global freshwater cycle, predicted to amplify with global warming; hence the importance of knowledge of this surface flux, of its impacts on the ocean, and of the ocean's redistribution (and storage) of these impacts. Salinity, in comparison, is of indirect interest, for its role in seawater density and buoyancy, etc. Bacon et al. (2015) recognize that a surface flux requires definition of a surface area." and "Since this is a review of existing literature and in light of established practice, we continue to employ here the "traditional" approach to freshwater flux calculation by use of a fixed reference salinity, but we note that – if knowledge of the surface atmosphere-ocean freshwater is required – then the approach of Bacon et al. (2015) is preferable."

We note that we already answered this point in response to Ursula Schauer's comment in the first round of reviews. This manuscript is not about creating new knowledge, it is about reporting on what has been published. And what has been published uses this concept.

8. Li 58: "the concept of" instead of "the existence of such a concept as".
   The sentence has been changed as suggested.

9. Li 60: remove prime after freshwater-
   The sentence has been changed as suggested.

10. Li 99: Here OK to mention the two different contributions from GIS but could also mention its relatively small contribution in the Arctic freshwater budget and comment on the location and rapid export through Fram Strait.
    We have added a sentence mentioning this "Given this region is adjacent to the Fram and Nares Straits, it's likely much of the discharge from northern Greenland is rapidly exported." and we have also clarified the section on glacier discharge to quantify Greenland melt and solid discharge on a regional basis.

11. Li 102-106: "Halocline" (or 'cline) concept in ocean does not only apply to the depth range there is a gradient between only two water masses. This statement from Rudels is misleading. Halocline is a depth range with strong salinity gradient (no matter how many water masses are involved). Please remove such comparisons with the "rest of the ocean". To call the halocline "a cold and freshwater mass" is not accurate- please reword.
    These sentences have been removed.

12. Fig 1 caption: Please also spell out VS and SZ.
    "; VS, Vilkiltskiy Strait; SZ, Severnaya Zemlya" has been added to the end of the Figure 1 caption.

13. Fig 2A y-axis label is missing. Fig 2C, please mark the BG region as defined.
    Figure 2C and Figure 2A caption changed as suggested.

14. Fig 3B. Please mark BG. Unit of FWC anomaly is in km^3. Fig 2c in m. Satellite observations: It seems like you cut across the Arctic Ocean along 60E. There must be more data on the "other hemisphere", particularly important for the freshwater budget. Table 2. It would help to show the time and space distribution of the new data in a figure. Table could include total number of profiles. Where is NABOS data?
    The units in Fig 2b and c are correct, but the label "FWC" is not. The figure shows freshwater inventories, in m (fraction of the water column representing pure freshwater). The y-axis label has been adjusted to read: "FW inventory difference (m). Likewise for Fig 3b, y-axis label now reads "FW inventory anomaly (m)".
    We apologize for the omission and thank the reviewer for pointing this out. NABOS data are included up to 2013; the NABOS 2015 data is not included in the analysis. We have augmented

Table 2 to that effect. All other data is listed in Rabe et al. (2014). We have added a figure to Figure 3 to show all data points used in the objective analysis. Time is shown in color of each of the location dots, making the data coverage visible, in particular for the later years. For further detail on earlier years, up to 2012, please refer to a similar plot in the Supplement to Rabe et al. (2014).

15. Li 240: require[s]
Text changed as suggested.

16. Li 261: delete "and changes in"
Text changed as suggested.

17. Li 274: increasing rain as climate warms- any citation on this for the Arctic Ocean?
A publication by Bintanja (2018) has been added.

18. Li 290: instead of "up to 50%" please give a range from ORAs.
Our apologies for this text, it is not relevant to describe Figure 5. We have changed the sentence to read, "Freshwater stored in sea ice, i.e., sea ice volume, decreased by roughly 10% for maximum sea ice and 40% for minimum sea ice over 2000-2010 (Figure 5)."

19. Li 300: You mean "New" sea ice in the Arctic forms on shelves?
We consider it is a tautology but added the word "new" nonetheless; the copy editor can remove it at a later stage.

20. Li 308: comment why faster TD matters, by exporting more FW?
We added the comment "thus capable of rapidly transporting sea ice out of the Arctic"

21. Li 327: in response to storms vertical mixing can also bring up oceanic heat and melt sea ice. There are several observations in the literature. Do you want to also include this in the "sea ice" section? [and/or in the vertical redistribution discussion?]
Following the suggestion, we have added discussions about storm-driven mixing and its impacts on sea ice melt and vertical distribution of freshwater at the end of the first paragraph in Sea Ice section and the third paragraph in Redistribution of Arctic Freshwater section. To support the discussions, we have also added two most recent citations.

22. Page 18: if so important, please mark the mentioned glaciers in Fig 1 so the reader sees where they are relative to the Arctic Ocean.
Thank you for this comment. We have opted to just remove the glacier names.

23. Li 380-395: around here, more comments are needed on the location of Greenland in the large Arctic Ocean freshwater landscape and eventually to tone down its contribution. So, how much km^3 of FWC is actually going into the Arctic Ocean freshwater balance?
We have edited the whole section for clarity and added the regional breakdown of ice loss from northern Greenland that goes directly into the Arctic Ocean, as well as the contribution from other glaciated regions within the Arctic including Svalbard, Arctic Canada and the Russian high Arctic archipelago.

24. Li 404: define mSv
mSv changed to milli-Sverdrup and we added the specification (1 Sv = 10^6 m3 s-1).

25. Li 436-437: an entire sentence is repeated.
Repeated sentence removed.

26. Section 3 is not well structured. It is a mixture of different processes and regions and their role in redistribution (lateral and vertical). There is also some overlooked literature and processes.

We have restructured Section 3 by adding subsections on; Competing processes in the Beaufort Gyre, Wind-driven circulation, Ekman pumping and wind mixing, Geostrophic density driven flow and shelf cascading.

Comments from Reviewer #1:

**Major comments**

The authors have softened their claim that there is fresh water compensation between the Beaufort Gyre and the rest of the Arctic in the abstract, however the presentation throughout the manuscript still needs significant improvement. Studies that suggest compensation seem to be preferentially cited (See L162 comment below) and some seem to be misrepresented (see L159 comment).

The analysis of the reanalysis products is much more comprehensively presented, but it would be very helpful to include boxes indicating which areas are being integrated over in addition to describing the regions in the Figure 2 caption. The same holds for Figure 3.

The regions in Figure 3 are now marked.

The satellite and in-situ fresh water estimates are very different during the period that this study focuses on, yet this is not discussed. The methods used for the satellite calculation are not presented (what empirical constants are used? What isohaline/reference salinity is being targeted?), so it is unclear how we might expect them to compare with in-situ measurements.

Details about the satellite calculations are now included in the text.

Compensation in the satellite estimate appears to decrease in the last part of the record. This is also not discussed. It is unclear why half of the Arctic is left out of (part of?) this analysis, and also unclear why an insitu estimate for the Beaufort Gyre is not presented. Presumably it is a subset of the full Arctic dataset? Please explain these analysis choices.

It is well-known that, while the sea surface height variability in the BG region is dictated by the variability in salinity, the same in the Nordic Seas and the Barents Sea is controlled by Atlantic Water temperature in comparison to salinity (Raj et al., 2021). Hence the methodology to estimate FWC from sea surface height data is not recommended in those two regions. Our study included the rest of the Arctic excluding the Canadian Archipelago, Nordic and the Barents Sea.

My recommendation is for major revisions because these observational analyses are a foundational part of this study and it is important that they be presented clearly, but note that the changes are much less significant than those recommended in the last revision.

**Specific Comments**

*Note: line numbers refer to the tracked-changes version of the manuscript.*

1. L67: Generally deltaS=S_ref-S, please check and revise.

   We have changed the sentence to read, "It usually manifests as a small fraction of the seawater volume or flux, where the fraction takes the form ($\delta S/S_{ref}$), and where $\delta S = S–S_{ref}$ is the deviation of the seawater salinity $S$ from a reference value $S_{ref}$, and where the sign in the numerator is conventionally reversed, so that a positive scaled salinity anomaly reflects a freshwater reduction, and vice-versa."

2. L131: Only Greenland ice sheet contributions are discussed, this could be a good place to also introduce GIS runoff contributions.

   We have added further detail on the sources of runoff from Greenland in this section.

3. L139: Not sure that adding the depths of the halocline and Atlantic layer is useful, as they vary substantially across the Arctic.

   We removed the depth information.

4. L159: Which studies are you referring to? (Proshutinsky et al., 2009; McPhee et al., 2009; Rabe et al., 2011; Haine et al., 2015)? It is inaccurate to state that these studies do not take redistribution or the Greenland Ice Sheet into account.
Thank you for this comment. We have revised the sentence to correct this inaccurate statement.

5. L161: Remove "(among other processes)" or explain which processes you are referring to.
"Among other processes" removed as suggested.

6. L162: Rabe et al. 2011 and Morison et al. 2012 do not quantify the degree of compensation between the Beaufort Gyre and the rest of the Arctic Ocean. Wang et al. 2019 "Recent Sea Ice Decline Did Not Significantly Increase the Total Liquid Freshwater Content of the Arctic Ocean", show an updated time series from Rabe, which suggests that there has not been compensation between the Amerasian and Eurasian basins.
Thank you for pointing this out. The sentence is not fully correct as is. We have deleted the sentence, as redistribution of freshwater is discussed in detail in Section 3.".
Wang et al. (2019) show timeseries of liquid freshwater content (FWC) based on observational analysis for: the whole Arctic deep basin (down to 34 isohaline); the Amerasian Basin; and the Eurasian Basin. Wang et al. argue that most of the changes in liquid FWC occurred in the Amerasian Basin, they don't conclude from the observational timeseries that there is no compensation between the Beaufort Gyre and the remaining Arctic. The Amerasian Basin encompasses a significantly larger area than the Beaufort Gyre typically covers.
Indeed, Rabe et al. (2011), showing a comparison between two time periods, do actually not conclude that there is a full compensation between the Beaufort Gyre and the remaining Arctic Ocean basin. Rather, they state that there is a general increase in FWC, accompanied by a decrease in specific regions (e.g. the Eurasian Basin north of the Laptev Sea). They also conclude from an ice-ocean model simulation that the overall liquid freshwater content change is related to an Ekman Pumping in the Amerasian Basin.

7. Figure 2: It would be helpful to show which areas are summed over to produce panel D in panel C.
The area used to define the average over the Beaufort Gyre is now shown in Figure 2c.

8. L226: It would be useful to provide a brief review of the method used in Giles et al. 2012 and Armitage et al. 2016, and now in this paper, including the empirical constant used, for clarity.
We have added this text to include a review of the Giles et al. (2012) and Armitage et al. (2016) methods, "It derives from the perceptions (1) that the sea surface height change (as observed by satellite altimeters) is the sum of two components: mass addition (or loss) and steric expansion (or contraction); and (2) that observation of mass changes (by satellite gravimetry) enables separation of these two components. A two-layer model is assumed, where the sea surface height and interface depth are variable, where the upper layer represents the halocline (and surface mixed layer) and the lower layer all underlying waters. Upper layer thickness changes (per unit water column area) are then a function of changes in sea surface height and water column mass, with assumed layer densities; and changes in freshwater content are then the thickness changes scaled by ($\delta S/S_{ref}$), with reversed sign. Giles et al. (2012) assume $S_{ref}$ (their $S_2$) = 34.7 and upper layer salinity 27.7. While this is a very simple model, the observed signals are significantly larger than the uncertainty, as shown in their thorough uncertainty assessment (Giles et al. 2012 Supplementary Information)."

9. L265: Why aren't these areas included in Figure 3? Please explain this further.

It is well-known that, while the sea surface height variability in the BG region is dictated by the variability in salinity, the same in the Nordic Seas and the Barents Sea is controlled by Atlantic Water temperature in comparison to salinity (Raj et al., 2021). Hence the methodology to estimate FWC from sea surface height data is not recommended in those two regions. Our study included the rest of the Arctic excluding the Canadian Archipelago, Nordic and the Barents Sea.

The northward extent of the state-of-the-art altimeter SSH data developed under the ESA CCI SLBC project is 82.5N. The 'polar gap' is still a major issue. There are ongoing projects (ESA funded CRYO-TEMPO) which is trying to address this issue.

10. L278: Does this mean that the in-situ estimates at the end of the record may be underestimated? In this case, the discrepancy between the satellites and in-situ estimates may be even larger than what is presented here.

The mapping procedure uses a moving window of +- 3 years from the month of mapping. Those monthly maps are then annually averaged to obtain the liquid freshwater content. As the main increase in liquid freshwater occurred between , we cannot say if 2015 is underestimated, overestimated or the same as if we had used data up to 2018. We have adjusted the text to read: "Due to the method the annual values are biased towards the prior three years near the end of the timeseries, as the mapping analysis only includes data up to 2015; and 2012, 2013 and 2014 show similar levels as 2015. This could lead to an under- or overestimate of the annual mean value for 2013, 2014 and 2015."

11. Figure 3: Please indicate where your Beaufort Gyre region in panel B. In the caption you state that the time series are calculated to the 34 isohaline using a reference salinity of 35. This is likely not the case for the satellite calculation, which uses an empirical constant to relate steric changes to freshwater changes and should be clarified. Again, why is the in-situ BG time series not shown?

The bold black frame in the revised Figure 3b indicates the location of the BG. Details regarding the estimation of satellite derived FWC listed above in major comments.

12. L329: Please justify this statement and include a citation: "Precipitation over the Arctic is the main source of freshwater into the Arctic Ocean", including that from river discharge from the large continental drainage basins".

The total continental runoff into the Arctic Ocean is about 0.1 Sv (see Table 1, Haine et al., 2015). The remaining sources are lower, those of similar order of magnitude are Precipitation-Evaporation and Bering Strait liquid inflow. According to Zhang et al. (2013), river discharge from the surrounding continental drainage basins is predominantly contributed to by precipitation, though land surface processes (e.g., thawing permafrost, decreasing vegetation transpiration) may have small contributions. So, the total precipitation directly falling down to the Arctic Ocean and sea ice surface and indirectly through river discharge is much larger than the oceanic freshwater input through the Bering Strait. We have revised the sentence in the manuscript with more detailed discussions on this.

We have added appropriate citations (Haine et al., 2015; Serreze et al., 2006).

13. L456: Could reconsider the use of the word "Discharge" here as it is usually associated with only the solid component and you also discuss liquid runoff.

As the most recent studies explicitly call their data product discharge (see e.g. Mankoff et al., 2019) we prefer to keep the term, but we have added a line to make it more explicit that we include both solid and liquid discharge in this term.

14. L497: It would be useful if you could explain and lay out why the total flux is larger than the net mass loss in a simpler fashion.

We have revised the whole section and made it shorter and simpler to read and included a clearer explanation of the difference between total flux and net mass loss.

15. L509: Does Mankoff consider only solid freshwater flux? Or total? Please clarify.

Mankoff considers both solid and liquid, hence why we prefer to keep the term. We have explicitly stated this in the text.

16. L544: Translate Gt to mSv to make comparable with other sources.

Thank you, we appreciate this comment but have chosen to keep Gt to be consistent with the source references.

17. L559: Could remove "as we do" since you are actively noting it in that sentence.

Wording "as we do" removed.

18. L628: Odd phrasing. Steric height changes do not redistribute fresh water, rather they are reflective of freshwater redistribution.

Steric effects change the sea surface height, which in turn drives barotropic geostrophic currents (here ignoring the effect of internal density gradients adding to the total geostrophic flow).

We added "by driving near-surface geostrophic currents"

19. L640: Sentence is repeated.

Repeated sentence removed.

20. L696: I didn't follow the last clause of this sentence, please clarify.

We split the sentence in two so that we could more clearly explain the last clause:

Further, Liang and Losch (2018) show not only that the positive feedback loop "enhanced vertical mixing = less sea ice" reduces the halocline-to-Atlantic Water (AW) stratification, but also leads to a colder AW. They argue that this colder AW combined with enhanced vertical mixing leads to an increased mixing between AW and deeper water masses, thus bringing more salt from AW to the deep Arctic.

21. L706: Suggest weakening "is due" to "is potentially due", or "appears to be".

We feel "is due in part" is sufficiently weakened.

22. L714: Unclear why "to distinguish trends from low frequency variability" is included here.

Here we meant that continuing monitoring would help distinguish climate change forced long-term trends from low frequency variability. The sentence in the manuscript has been reworded.

Comments from Reviewer #2

1. Line 105: The statement that in the Arctic "the halocline is a cold and fresh water mass" is not that accurate. As everywhere, the Arctic halocline is still dominated by strong vertical salinity gradient (e.g. it is not fresh). And the composition of Arctic halocline is not simple. In the Eurasian Basin, halocline is composed of cold halocline layer (where temperature is close to the freezing point and salinity increases with depth) and lower halocline water (where both temperature and salinity rapidly increase with depth). In the Amerasian Basin, this structure is further complicated by the presence of Pacific waters.
   We have deleted the wording, "As noted by Rudels et al. (2004), the term "halocline" is misleading yet common practice. In the rest of the world ocean, halocline denotes the depth range where salinities abruptly change as two water masses mix; in the Arctic, the halocline is a cold and fresh water mass.".

2. Line 305: I am not sure that one-year long travel time from the Laptev Sea to Fram Strait is a realistic estimate.
   During MOSAiC, ice formed north of the New Siberian Islands in December 2018, then reached the Fram Strait in late spring 2020 (Krumpen et al., 2020; The Cryosphere, 14, 2173–2187, 2020; https://doi.org/10.5194/tc-14-2173-2020 ). This was an extreme year, but demonstrates that a 1.5 year travel time is realistic. We have changed the wording to "1.5-3 years".

3. Section 2.4. Even though this section is mostly about the Greenland Ice Sheet, the contribution to the Arctic FW balance comes (and is discussed) from the Greenland and other glaciers. So, the title may be changed to better reflect these various sources of FW. I found that this part is disproportionally long relative to other sections. A lot of discussion which is not directly relevant to the Arctic FW can be omitted. I recommend to compare contributions of different sources (PE, sea ice, riverine water, including Greenland and glacier ice melt) to the overall Arctic FW (a summary Table?). E.g. there is no direct comparison of the Greenland ice to the Arctic FW balance. One can deduce it from the text comparing 26Gt versus sea ice decline from Fig 5, for example. But it will be beneficial for the paper to have such a summary.
   We have thoroughly revised this section and renamed it "Freshwater Flux Discharge from glaciers and the Greenland ice sheet" including adding in some context from other glaciated parts of the Arctic that also contribute freshwater and some more easy to compare numbers.

4. Line 393: "where the identify" needs edits.
   This section has been changed to read, "The Bamber study estimates that including land ice from other parts of the Arctic as well as the Greenland ice sheet, the total freshwater flux is around 1300 Gt per year in the period since 2010. They also identify a marked increase in runoff and discharge compared to a climatology period of 1960 to 1990."

5. I agree with the other reviewer, that the paper (still) lacks comprehensive discussion of spatial redistribution of FW within the Arctic Ocean proper.
   Please see Review #3 comment #12. We have added the suggested text about redistribution for the 2010-2019 period and the relation to atmospheric variability to the manuscript.

Comments from Reviewer #3 (Jamie Morison)

This is a review of "Freshwater in the Arctic Ocean 2010-2019" by Solomon and others. The authors review relevant literature and make their own computations of freshwater content (FWC) for the Arctic Ocean from hydrography and remote sensing (altimetry and GRACE). They argue that Arctic Ocean freshwater content has stabilized since 2010. The authors have responded to prior review comments and the revised paper is a substantial improvement over the earlier version. However, the authors themselves raise questions about adequacy of their data for reaching solid conclusions. As they point out, inadequate spatial coverage is a likely reason freshwater content from their in-situ observations do not agree with their remote sensing after 2010.

Further, although they are using altimetry-derived sea surface heights from DTU that they argue uses a retracker suitable for detecting sea surface height in sea ice regions, they have no remote sensing-derived freshwater content in major important regions: north of 82°N and the whole Fram Strait quadrant of the Arctic Ocean. Therefore, it isn't clear from the data analysis what freshwater content has done in the 2010s. Consequently, as it stands the utility of the paper is its timely review of research. Additional references suggested below or a closer look at existing ones might help improve the data analysis and provide additional insights into changing freshwater content.

We thank the reviewer for pointing this out. We agree that there are few points not been made clear in the previous versions of the manuscript. The spatial distribution of the ESA CCI SLBC SSH data produced by DTU, used in this study is shown below (left panel; Figure from Raj et al., 2020). Note that the latitudinal extent of the data is 82.5N. In the manuscript Figure 3b (right panel shown below) shows the region in which the FWC is calculated from satellite data. The Nordic Seas and the Beaufort Gyre is purposefully excluded because the sea surface variability in these regions are dominated by the changes in Atlantic Water temperature in comparison to salinity changes (Raj et al., 2020). Hence the methodology used (detailed in the revised version) may not be accurate in these two regions, in comparison to BG region where SSH variability is due to changes in salinity (e.g., Raj et al., 2020).

[Figure]

-Best regards,
Jamie Morison

Detailed Comments

1. L130 and Captions Figure Labels Fig. 2a - Units? Freshwater content in meters? What are the offsets for the different lines?
   Figure 2a units were cropped out and the figure has been corrected. The offsets are due to different FWC estimates in each model.

2. L132-137 - This is an important point. Fig. 2b shows the increase in FWC in the Beaufort Sea is compensated at least to some degree by FWC loss on the Russian- Eurasian side of the Arctic Ocean. This is characteristic of the cyclonic mode of circulation [Morison et al., 2021; Morison et al., 2012; Sokolov, 1962]. The wide variability in FWC change among the ORAs on the Russian side is likely due the paucity of observations there in recent years. Morison et al., (2021) speculate that the in situ observations have had an increasing spatial bias toward the Beaufort Sea. It would be a great contribution if this paper could address this by somehow illustrating the locations observations that went into the ORAs.
   Thank you very much for these comments. The text now reads, "This is characteristic of the cyclonic mode of circulation (Morison et al., 2021; Morison et al., 2012; Sokolov, 1962). However, there is a significant spread in estimates of freshwater content in the Beaufort Gyre and the rest of the Arctic Ocean (Figure 2d), which prevents a definitive estimate of the degree of this compensation. The wide variability in FWC change among the ORAs on the Russian side is likely due the paucity of observations there in recent years. Morison et al. (2021) speculate that the in-situ observations have had an increasing spatial bias toward the Beaufort Sea. This highlights the need to be able to estimate the redistribution of freshwater when assessing changes in Arctic Ocean freshwater, as well as the recent reduction in total Arctic Ocean freshening relative to the 2000-2010 period."

   The ORAs used here do not share datasets and the assimilation schemes treat data in different ways. Locations of assimilated observations are different among products because the original observational datasets may differ and the quality check criteria that are applied to data are also different. The distribution of all in-situ datasets used in this set of ORAs cannot be presented in the paper, but the following figures show the data distribution for the period 2004-2017 as given to the C-GLORS system (please note that Argo were not used before 2003). Assimilated data in other reanalyses might be similarly distributed but not exactly collocated.

[Figure]

3. L166-169 - Yes, but sea ice was one of the main motivations for launching CryoSat2, and there are efforts in addition to the ones used in this study that discuss dynamic ocean topography from CryoSat-2 for the ice-covered seas, e.g., Kwok and Morison, (2016, 2017); Armitage et al.( 2018a, 2018b).
We thank the reviewer for pointing this out. We agree that the statement above may not be correctly postulated. We have revised the text as follows:
"CryoSat-2, launched in 2010, is a satellite altimeter that provides coverage up to 88°N with much better spatial resolution than before. Several studies have utilized this source to study the sea level variability of the Arctic (Kwok and Morison, 2016, 2017; Armitage et al., 2018a, 2018b; Rose et al., 2019; Raj et al., 2020). However, constructing precise altimeter derived sea level data in the Arctic Ocean is still a challenge, one of them is the effect of melt ponds during summer on the waveforms which dominate the reflected signal. A better understanding of the radar altimeter response over the different ice types must be gained to improve the quantity and quality of the range retrievals in the Arctic Ocean. One of the ongoing efforts is the currently ongoing CRYO-TEMPO project, funded by the European Space Agency."

4. L170-179 - Why is there no DTU satellite data north of 82°N or in the Fram Strait quadrant? The various versions of the DTU mean sea surface (MSS) don't have gaps this large. These are critical regions, particularly the Eastern Arctic, where we think in situ observations may be lacking.
The northward extent of the state-of-the-art altimeter SSH data developed under the ESA CCI SLBC project is 82.5N. The 'polar gap' is still a major issue. There are ongoing projects (ESA funded CRYO-TEMPO) which is trying to address this issue.

5. L180-189 - It seems like a comparison with the cyclonic mode of Morison et al. (2021) would be appropriate here. For 2010-2019, their results suggest a brief initial shift out of the cyclonic mode in response to a record low winter AO index in 2010 that tended to return freshwater to the Russian side of the Arctic Ocean. This was followed by a return to the cyclonic mode for most of the decade tending to shift freshwater once again out of the East longitudes and into the Beaufort Sea. The mix of circulation regimes would make it hard to draw a general conclusion about FWC change in the 2010s, especially not having regular repeat in situ observations well distributed across the Arctic Ocean.
   We include a brief discussion of these results as a response to your comment #12.
6. L199-201 - Yes, I agree "a good part of the difference after 2009 may be due to the contribution by the rest of the basin outside the Beaufort Gyre."
   In situ observations outside the Beaufort Gyre have become increasingly rare.
   In a review it would be impactful to illustrate this with a figure showing the locations of the hydrographic stations used in the analysis.
   We have added a map of station locations used for the objective analysis to Figure 3. A subset of this map can be found in the supplement to Rabe et al. (2014).
7. For the original research part of this paper, a good way to combine remote sensing and in situ observations is to first compare them say for FWC (or for altimetry derived DOT and in situ derived dynamic heights [Morison et al., 2018; Morison et al., 2012]) at the locations and times of the in-situ observations. Then after making reasonable corrections, apply the remote sensing FWC to the whole Arctic Ocean and for other times with some confidence that these remote sensing results represent a good proxy for in situ observations [Morison et al., 2018; Morison et al., 2012]. The authors could do this by combining the data in Table 2 and the DTU-derived FWC. It would also be illuminating and relatively easy to compare DTU-derived FWC to the FWC of the ORAs and see how the regions of good and bad comparison relate to the distribution of in situ observations.
   We appreciate this suggestion and it would be a very interesting and insightful study but have decided it is beyond the scope of this review paper.
8. Table 2 - Why are there no NABOS data? Except for perhaps the Polarstern data, there isn't much data at all along the Russian margins of the Basin where Rabe et al. (2011) and Morison et al (2012) found freshwater decrease offsetting the Beaufort Gyre increases in the 2000s. Again, it would be helpful to show charts of the data locations to illustrate if there is a geographic sampling bias.
   We apologize for the omission and thank the reviewer for pointing this out. NABOS data are included up to 2013; the NABOS 2015 data is not included in the analysis. We have augmented Table 2 to that effect. All other data is listed in Rabe et al. (2014). As we do not want to duplicate information in a publication, we only include a comprehensive source list in Table 2.
9. L260 - Figure 4 is an analysis by the authors, not Thompson and Wallace, correct? In Figure 4, the positive correlation after 2010 is clear but the negative correlation before is not so obvious.
   "Figure 4" has been removed from the sentence to prevent misunderstanding.
10. L383 - The Hill et al. reference is missing.
    The reference has been added to the paper:

Hill, E. A., Carr, J. R., Stokes, C. R., and Gudmundsson, G. H.: Dynamic changes in outlet glaciers in northern Greenland from 1948 to 2015, The Cryosphere, 12, 3243–3263, https://doi.org/10.5194/tc-12-3243-2018, 2018.

11. L385 - The glaciers drain into the shelf regions around Greenland and are carried south, so their contribution to Arctic Basin is undoubtedly very small. It is hard to see the point in considering Greenland melt in the freshwater balance of the Arctic Ocean, even more so considering the complete lack of FWC data shown for the region around Northern Greenland. The freshwater flux from Greenland has been estimated by models and satellite derived estimates of ice velocity but it is true that the flux is small into the Arctic Ocean compared to other contributions. However, this flux has been increasing and will likely increase further in coming decades. We have made this point more explicit in the text and we have thoroughly revised the section to make it more concise. We have also renamed the section and included figures for other glaciers in the central Arctic to set it in context.

12. L424 and Redistribution of Arctic Freshwater section generally - The results of Morison et al. (2021) provide a longer-term perspective on the effect on freshwater distribution of the cyclonic mode of circulation epitomized in the changes of the early 90s (Morison et al., 2000) and 2007-2008 (Morison et al., 2012). The cyclonic mode is characterized as the first EOF of ocean surface height variability (dynamic heights, 1959-1989 and satellite DOT, 2004-2019). In the cyclonic phase, surface depression, reduced freshwater content, and cyclonic circulation are imposed on the whole Russian side of the Arctic Basin. This offsets the gains in freshwater in the Beaufort Gyre. The cyclonic mode is related to the AO index with order 1-year latency, and has become more prominent under enhanced winter AO since 1990.
These comments have been added to the end of the first paragraph of Section 3.
We agree on the comment that the cyclonic mode is also important in additional to the conventionally emphasized Beaufort Gyre. We have therefore added the text following the reviewer's comment in the frist paragraph in the section of Redistribution of Arctic Freshwater.

13. L464-466 - Dewey et al. (2018) indicate the Ekman pumping and ice drag feedback mechanism stabilize ice and ocean velocity at time scales of less than a week while eddy propagation feedback has time scales measured in years.
Thank you for this comment. This sentence has been added to the text.

References

Armitage, T. W. K., S. Bacon, and R. Kwok (2018a), Arctic Sea Level and Surface Circulation Response to the Arctic Oscillation, Geophysical Research Letters, 45(13), 6576-6584, doi:10.1029/2018GL078386.

Armitage, T. W. K., R. Kwok, A. F. Thompson, and G. Cunningham (2018b), Dynamic topography and sea level anomalies of the southern ocean: Variability and teleconnections, J. Geophys. Res., 123(1), 613−630.

Kwok, R., and J. Morison (2016), Sea surface height and dynamic topography of the ice-covered oceans from CryoSat-2: 2011-2014, J Geophys. Res. - Oceans, 121, 674-692,

doi:10.1002/ 2015JC011357.

Kwok, R., and J. Morison (2017), Recent changes in Arctic sea ice and ocean circulation, US CLIVAR Variations: Summer 2017 15(3), 1-6.

Morison, J. H., K. Aagaard, and M. Steele (2000), Recent environmental changes in the Arctic: A review, Arctic, 53(4), 359-371.

Morison, J., R. Kwok, S. Dickinson, R. Andersen, C. Peralta-Ferriz, D. Morison, I. Rigor, S. Dewey, and J. Guthrie (2021), The Cyclonic Mode of Arctic Ocean Circulation, Journal of Physical Oceanography, 51(4), 1053-1075, doi:10.1175/JPO-D-20-0190.1.

Morison, J., R. Kwok, S. Dickinson, D. Morison, C. Peralta-Ferriz, and R. Andersen (2018), Sea State Bias of ICESat in the Subarctic Seas, IEEE Geoscience and Remote Sensing Letters, 15(8), 1144-1148, doi:10.1109/LGRS.2018.2834362.

Morison, J. H., R. Kwok, C. Peralta-Ferriz, M. Alkire, I. Rigor, R. Andersen, and M. Steele (2012), Changing Arctic Ocean freshwater pathways, Nature, 481(7379), 66-70, doi:http://www.nature.com/nature/journal/v481/n7379/abs/nature10705.html - supplementary-information.

Sokolov, A. L. (1962), Drift of ice in the Arctic Basin and changes in ice conditions over the northern sea route, Probl. Arct. Anarct. Engl. Translation, 11, 1-20.

---

## Author Response (AR3)

The Editor has done an excellent job editing this manuscript and guiding the review process. On behalf of all of the authors, thank you for your careful and thorough work. We have revised the manuscript to address the list of technical corrections in your final review.

All the best,

Amy Solomon